# DELVE: feature selection for preserving biological trajectories in single-cell data

Jolene S. Ranek[1,2], Wayne Stallaert [3], J. Justin Milner [4,5], Margaret Redick[1,2], Samuel C. Wolff[1,2], Adriana S. Beltran[1,6], Natalie Stanley [2,7] & Jeremy E. Purvis [1,2]

Single-cell technologies can measure the expression of thousands of molecular features in individual cells undergoing dynamic biological processes. While examining cells along a computationally-ordered pseudotime trajectory can reveal how changes in gene or protein expression impact cell fate, identifying such dynamic features is challenging due to the inherent noise in single-cell data. Here, we present DELVE, an unsupervised feature selection method for identifying a representative subset of molecular features which robustly recapitulate cellular trajectories. In contrast to previous work, DELVE uses a bottom-up approach to mitigate the effects of confounding sources of variation, and instead models cell states from dynamic gene or protein modules based on core regulatory complexes. Using simulations, single-cell RNA sequencing, and iterative immunofluorescence imaging data in the context of cell cycle and cellular differentiation, we demonstrate how DELVE selects features that better define cell-types and cell-type transitions. DELVE is available as an open-source python package: https://github.com/jranek/delve.

High-throughput single-cell technologies, such as flow and mass cytometry[1–3], single-cell RNA sequencing[4–7], and imaging-based profiling techniques[8–11] have transformed our ability to study how cell populations respond and dynamically change during processes like cellular differentiation[12–16], cell cycle[17–19], and immune response[20–22]. By profiling many features (e.g., proteins or genes) for many thousands of cells from a biological sample, these technologies provide high-dimensional snapshot measurements that can be used to gain fundamental insights into the molecular mechanisms that govern phenotypic or pathological changes.

Trajectory inference methods[23] have been developed to model dynamic biological processes from snapshot single-cell data. By assuming cells are asynchronously changing over time such that a profiled biological sample from a single experimental time point describes a range of the underlying dynamic process, computational trajectory inference approaches have leveraged minimum spanning tree approaches[24–26], curve-fitting[27,28], graph-based techniques[12,29,30], probabilistic approaches[31–33], or optimal transport[34,35] to order cells based on their similarities in feature expression. Once a trajectory model is fit, regression[36–38] can be performed along estimated pseudotime (e.g., distance through the inferred trajectory from a start cell) to identify specific cell state changes associated with differentiation or disease trajectories. Moreover, these inferred cellular trajectories have the potential to elucidate higher-order gene interactions[39], gene

[1]Department of Genetics, University of North Carolina at Chapel Hill, Chapel Hill, NC, USA. [2]Computational Medicine Program, University of North Carolina at Chapel Hill, Chapel Hill, NC, USA. [3]Department of Computational and Systems Biology, University of Pittsburgh, Pittsburgh, PA, USA. [4]Department of Microbiology and Immunology, University of North Carolina at Chapel Hill, Chapel Hill, NC, USA. [5]Lineberger Comprehensive Cancer Center, University of North Carolina at Chapel Hill School of Medicine, Chapel Hill, NC, USA. [6]Human Pluripotent Cell Core, University of North Carolina at Chapel Hill School of Medicine, Chapel Hill, NC, USA. [7]Department of Computer Science, University of North Carolina at Chapel Hill, Chapel Hill, NC, USA. ✉e-mail: natalies@cs.unc.edu; purvisj@email.unc.edu

regulatory networks[40], predict cell fate probabilities[32], or find shared mechanisms of expression dynamics across disease conditions or species[41,42].

While trajectory analysis has proven useful in the context of single-cell biology, the identification of characteristic genes or proteins that drive continuous biological processes relies on having inferred accurate cellular trajectories, which can be challenging, especially when trajectory inference is performed on the original full unenriched dataset. Single-cell data are noisy measurements that suffer from limitations in detection sensitivity, where dropout[43], low signal-to-noise, or sample degradation[44] can result in spurious signals that can overwhelm true biological differences. Furthermore, all profiled sources of feature variation contribute to the cell-to-cell distances that define the inferred cellular trajectory; thus, including confounding sources of biological variation (e.g., cell cycle, metabolic state) or irrelevant features (e.g., extracted imaging measurements that contain low signal-to-noise ratio) can distort or mask the intended trajectory of study[45,46]. With the accumulation of large-scale single-cell data and multi-modal measurements[47], appropriate filtering of noisy, information-poor, or irrelevant features can serve as a crucial and necessary step for cell type identification, inference of dynamic phenotypes, and identification of punitive driver features (e.g., genes, proteins).

Feature selection methods[48] are a class of supervised or unsupervised approaches that can remove redundant or information-poor features prior to performing trajectory inference, and therefore, they have great potential for improving the interpretation of downstream analysis, while easing the computational burden by reducing dataset dimension. In the supervised-learning regime, classification-based[49] or information-theoretic approaches[50,51] have been used to evaluate features according to their discriminative power or association with cell types. Despite having great power to detect biologically-relevant features, these methods rely on expensive or laborious manual annotations (e.g., cell types) which are often unavailable[52] thus precluding them from use. In the unsupervised-learning regime, computational approaches often aim to identify relevant features based on intrinsic properties of the complete dataset; however, these methods have some limitations with respect to retaining features that are useful for defining cellular trajectories. Namely, although unsupervised variance-based approaches[53,54], which effectively sample features based on their overall variation across cells, have been extensively used to identify features that define cell types without the need for ground truth

annotations, (1) they can be overwhelmed by noisy or irrelevant features that dominate data variance, and (2) are insensitive to lineage-specific features (e.g., transcription factors) that have a small variance and gradual progression of expression. Alternatively, unsupervised similarity-based[29,55,56] or subspace-learning[57,58] feature selection methods evaluate features according to their association with a cell-similarity graph defined by all features or the underlying structure of the data (e.g., pairwise similarities defined by uniform manifold approximation and projection (UMAP)[59], eigenvectors of the graph Laplacian matrix[60]). While these approaches have the potential to detect smoothly varying genes or proteins that define cellular transitions, they rely on the cell-similarity graph from the full dataset and can fail to identify relevant features when the number of noisy features outweighs the number of informative ones[61,62].

To address these limitations, we developed DELVE (dynamic selection of locally covarying features), an unsupervised feature selection method for identifying a representative subset of molecular features that robustly recapitulate cellular trajectories. In contrast to previous work[29,53,55–58], DELVE uses a bottom-up approach to mitigate the effect of unwanted sources of variation confounding feature selection and trajectory inference, and instead models cell states from dynamic feature modules that constitute core regulatory complexes. Features are then ranked for selection according to their association with the underlying cell trajectory graph using data diffusion techniques. We demonstrate the power of our approach for improving inference of cellular trajectories through achieving an increased sensitivity to detect diverse and dynamically expressed features that better delineate cell types and cell type transitions from single-cell RNA sequencing and protein immunofluorescence imaging data. Overall, this feature selection framework provides an alternative approach for uncovering co-variation amongst features along a biological trajectory.

## Results

### Overview of the DELVE algorithm

We propose DELVE, an unsupervised feature selection framework for modeling dynamic cell state transitions using graph neighborhoods (Fig. 1). Our approach extends previous unsupervised similarity-based[29,55,56] or subspace-learning feature selection[58] methods by computing the dependence of each gene on the cellular trajectory graph structure using a two-step approach. Inspired by the molecular events

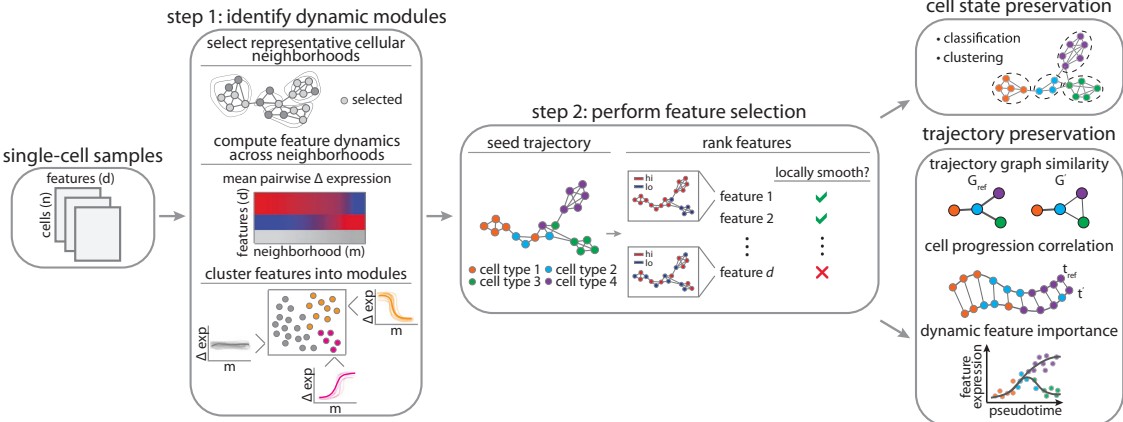

**Fig. 1 | Schematic overview of the DELVE pipeline.** Feature selection is performed in a two-step process. In step 1, DELVE clusters features according to their expression dynamics along local representative cellular neighborhoods defined by a weighted *k*-nearest neighbor affinity graph. Neighborhoods are sampled using a distribution-focused sketching algorithm that preserves cell-type frequencies and spectral properties of the original dataset[69]. A permutation test with a variance-based test statistic is used to determine if a set of features are (1) dynamically changing (dynamic) or (2) exhibiting random patterns of variation (static). In step 2, dynamic modules are used to seed or initialize an approximate cell trajectory graph and the trajectory is refined by ranking and selecting features that best preserve the local structure using the Laplacian Score[55]. In this study, we compared DELVE to the alternative unsupervised feature selection approaches on how well-selected features preserve cell type and cell type transitions according to several metrics.

that occur during differentiation, where the coordinated spatio-temporal expression of key regulatory genes govern lineage specification[63–66], we reasoned that we can approximate cell state transitions by identifying groups of features that are temporally co-expressed or co-regulated along the underlying dynamic process.

In step one, DELVE identifies dynamic modules of features that are temporally co-expressed by clustering features according to their average pairwise change in expression across prototypical cellular neighborhoods (See Fig. 1 Step 1, Step 1: dynamic seed selection). As has been done previously[67,68], we model cell states using a weighted $k$-nearest neighbor ($k$-NN) affinity graph, where nodes represent cells and edges describe the transcriptomic or proteomic similarity amongst cells according to all profiled features. Here, DELVE uses a distribution-focused sketching method[69] to effectively sample cellular neighborhoods across all cell types. This sampling approach has three main advantages: (1) cellular neighborhoods are more reflective of the distribution of cell states, (2) redundant cell states are removed, and (3) fewer cellular neighborhoods are required to estimate feature dynamics resulting in increased scalability. Following feature clustering, each module contains a set of features with similar local changes in co-variation across prototypical cell states along the cellular trajectory. Feature-wise permutation testing is then used to assess dynamic expression variation across grouped features as compared to random assignment. By identifying and excluding modules of features that have static, random, or noisy patterns of expression variation, DELVE effectively mitigates the effect of unwanted sources of variation confounding feature ranking and selection, and subsequent trajectory inference.

In step two, DELVE leverages modules of features with temporal co-expression variation to approximate the underlying cellular trajectory by constructing a new affinity graph between cells, where cell similarity is now redefined according to a core set of dynamically expressed regulators. All profiled features are then ranked according to their association with the underlying cellular trajectory graph using graph signal processing techniques[70,71] (See Fig. 1 Step 2, Step 2: feature ranking). More concretely, a graph signal is any function that has a real defined value on all of the nodes. In this context, we consider all features as graph signals and rank them according to their total variation in expression along the cellular trajectory graph using the Laplacian Score (LS)[55]. Intuitively, DELVE retains features that are considered to be globally smooth, or have similar expression values amongst similar cells along the approximate cellular trajectory graph. In contrast, DELVE excludes features that have a high total variation in signal, or expression values that are rapidly oscillating amongst neighboring cells, as these features are likely noisy or not involved in the underlying dynamic process that was seeded. The output of DELVE is a ranked set of features that best preserve the local trajectory structure. For a more detailed description on the problem formulation, the mathematical foundations behind feature ranking, and the impact of nonsense features on trajectory inference, see DELVE in the Methods section.

## DELVE outperforms existing feature selection methods in representing cellular trajectories in the presence of single-cell RNA sequencing noise

Although feature selection is a common preprocessing step in single-cell analysis[72] with the potential to reveal cell-type transitions that would have been masked in the original high-dimensional feature space[45], there has been no systematic evaluation of feature selection method performance on identifying biologically-relevant features for trajectory analysis in single-cell data, especially in the context of noisy data that contain biological or technical challenges (e.g., low total mRNA count, low signal-to-noise ratio, or dropout). In this study, we compared DELVE to eleven other feature selection approaches and evaluated methods on their ability to select features that represent cell

types and cell type transitions by performing two simulated single RNA sequencing studies where the ground truth was known. In the sections below, we will describe an overview of the feature selection methods considered and outline the simulation designs and evaluation criteria in more detail. We will then provide qualitative and quantitative assessments of how noise impacts feature selection method performance and subsequent inference of cellular trajectories.

**Overview of feature selection methods.** We performed a systematic evaluation of twelve feature selection methods for preserving cellular trajectories in noisy single-cell data. Methods were grouped into five general categories prior to evaluation: supervised, similarity, subspace-learning, variance, and baseline approaches. For more details on the feature selection methods implemented and hyperparameters, see Benchmarked feature selection methods and Supplementary Table 1.

**Supervised approaches.** To illustrate the performance of ground-truth feature selection that could be obtained through supervised learning on expert annotated cell labels, we performed Random Forest classification. Random Forest classification[49] is a supervised ensemble learning algorithm that uses an ensemble of decision trees to partition the feature space such that all of the cells with the same cell type label are grouped together. Here, each decision or split of a tree was chosen by minimizing the Gini Impurity score[73]. This approach was included to provide context for unsupervised feature selection method performance.

**Similarity approaches.** We considered four similarity-based approaches as unsupervised feature selection methods that rank features according to their association with a cell similarity graph defined by all profiled features (e.g., LS, neighborhood variance, hotspot) or dynamically-expressed features (e.g., DELVE). First, the Laplacian Score[55] is an unsupervised locality-preserving feature selection method that ranks and selects features according to (1) the total variation in feature expression across neighboring cells using a cell similarity graph defined by all features and (2) a feature's global variance. Next, neighborhood variance[29] is an unsupervised feature selection method that selects features with gradual changes in expression for building biological trajectories. Here, features are selected if their variance in expression across local cellular neighborhoods is smaller than their global variance. Hotspot[56] performs unsupervised feature selection through a local autocorrelation test statistic that measures the association of a gene's expression with a cell similarity graph defined by all features. Lastly, DELVE (dynamic selection of locally covarying features) is an unsupervised feature selection method that ranks features according to their association with the underlying cellular trajectory graph. First, features are clustered into modules according to changes in expression across local representative cellular neighborhoods. Next, modules of features with dynamic expression patterns (denoted as *dynamic seed*) are used to construct an approximate cellular trajectory graph. Features are then ranked according to their association with the approximate cell trajectory graph using the LS[55]. Given that DELVE is a model-free feature selection approach, we evaluated the robustness of DELVE by generating a distribution of accuracy scores across multiple runs of the method.

**Subspace learning approaches.** We considered two subspace-learning feature selection methods as unsupervised methods that rank features according to how well they preserve the overall cluster structure (e.g., multi-cluster feature selection (MCFS)) or manifold structure (e.g., SCMER) of the data. First, MCFS[58] is an unsupervised feature selection method that selects features that best preserve the multi-cluster structure of data by solving an L1 regularized least squares regression problem on the spectral embedding defined by all

profiled features. The optimization is solved using the least angles regression algorithm[74]. Next, single-cell manifold-preserving feature selection (SCMER)[57] is an unsupervised feature selection method that selects a subset of features that best preserves the pairwise similarity matrix between cells defined in uniform manifold approximation and projection[59] based on all profiled features. To do so, it uses elastic net regression to find a sparse solution that minimizes the KL divergence between a pairwise similarity matrix between cells defined by all features and one defined using only the selected features.

**Variance approaches.** We considered two variance-based feature selection approaches (e.g., highly variable genes[53], max variance) as unsupervised methods that use global expression variance as a metric for ranking feature importance. First, highly variable gene selection (HVG)[53] is an unsupervised feature selection method that selects features according to a normalized dispersion measure. Here, features are binned based on their average expression. Within each bin, genes are then z-score normalized to identify features that have a large variance, yet a similar mean expression. Next, max variance is an unsupervised feature selection method that ranks and selects features that have a large global variance in expression.

**Baseline approaches.** We considered three baseline strategies (e.g., all, random, dynamic seed) that provide context for the overall performance of feature selection. First, all features illustrate the performance when feature selection is not performed and all features are included for analysis. Second, random features represent the performance quality when a random subset of features are sampled. Lastly, dynamic seed features indicate the performance from dynamically-expressed features identified in step 1 of the DELVE algorithm prior to feature ranking and selection.

**Splatter single-cell RNA sequencing simulation study.** To validate our approach and benchmark feature selection methods on selecting genes that represent cellular trajectories, we simulated a total of 90 single-cell RNA sequencing datasets (1500 cells and 500 genes) with three trajectory structures (e.g., linear, bifurcation, tree) using Splatter. Here, for each trajectory structure, we generated 30 datasets. Splatter[75] simulates single-cell RNA sequencing data with various trajectory structures (e.g., linear, bifurcation, tree) using a gamma-Poisson hierarchical model. Importantly, this approach provides ground truth reference information (e.g., cell type annotations, differentially expressed genes per cell type and trajectory, and a latent vector that describes an individual cell's progression through the trajectory) that we can use to robustly assess feature selection method performance, as well as quantitatively evaluate the limitations of feature selection strategies for trajectory analysis. Moreover, to comprehensively evaluate feature selection methods under common biological and technical challenges associated with single-cell RNA sequencing data, we added relevant sources of single-cell noise to the simulated data. First, we simulated low signal-to-noise ratio by enforcing a mean-variance relationship amongst genes; this ensures that lowly expressed genes are more variable than highly expressed genes. Next, we modified the total number of profiled mRNA transcripts, or library size. This has been shown previously to vary amongst cells within a single-cell experiment and can influence both the detection of differentially expressed genes[76], as well as impact the reproducibility of the inferred lower-dimensional embedding[77]. Lastly, we simulated the inefficient capture of mRNA molecules, or dropout, by under-sampling gene expression from a binomial distribution; this increases the amount of sparsity present within the data. For more details on the splatter simulation, see Splatter simulation. For each simulated trajectory, we performed feature selection according to all described feature selection strategies, and considered the top 100 ranked features for downstream analysis and evaluation.

**Qualitative assessment of feature selection method performance.** Prior to evaluating feature selection method performance quantitatively, we began our analysis with a qualitative assessment of the importance of feature selection for representing cellular trajectories when the data contain irrelevant or noisy genes. First, we visually compared the cellular trajectories generated from a feature selection strategy with PHATE (potential of heat diffusion for affinity-based transition embedding). PHATE[78] is a nonlinear dimensionality reduction method that has been shown to effectively learn and represent the geometry of complex continuous and branched biological trajectories. As an illustrative example, Fig. 2a shows the PHATE embeddings for simulated linear differentiation trajectories generated from four feature selection approaches (all, DELVE, Laplacian Score (LS), and random) when subjected to a decrease in the signal-to-noise ratio. Here, we simulated a reduction in the signal-to-noise ratio and stochastic gene expression by modifying the biological coefficient of variation (BCV) parameter within Splatter[75]. This scaling factor controls the mean-variance relationship between genes, where lowly expressed genes are more variable than highly expressed genes (See Splatter simulation). Under low noise conditions where the data contained a high signal-to-noise ratio, we observed that excluding irrelevant features with DELVE or the Laplacian Score (LS) produced a much smoother, denoised visualization of the linear trajectory, where cells were more tightly clustered according to cell type. This was compared to the more diffuse presentation of cell states obtained based on all genes. We then examined how noise influences the quality of selected features from a feature selection strategy. As the signal-to-noise ratio decreased (high, medium, low), we observed that the linear trajectory became increasingly harder to distinguish, whereby including both irrelevant and noisy genes often masked the underlying trajectory structure (Fig. 2a all genes, medium to low signal-to-noise ratio). Furthermore, we found that unsupervised similarity-based or subspace learning feature selection methods that initially define a cell similarity graph according to all irrelevant, noisy, and informative genes often selected genes that produced noisier embeddings as the amount of noise increased (e.g., Fig. 2a LS: medium signal-to-noise ratio), as compared to DELVE (e.g., Fig. 2a DELVE medium signal-to-noise ratio). We reason that this is due to spurious similarities amongst cells, reduced clusterability, and increased diffusion times. These qualitative observations were consistent across different noise conditions (e.g., decreased signal-to-noise, decreased library size, increased dropout) and trajectory types (e.g., linear, bifurcation, tree) (See Supplementary Figs. 1–9). Although a qualitative comparison, this example illustrates how including irrelevant or noisy genes can define spurious similarities amongst cells, which can (1) influence a feature selection method ability to identify biologically-relevant genes and (2) impact the overall quality of an inferred lower dimensional embedding following selection. Given that many trajectory inference methods use lower dimensional representations in order to infer a cell's progression through a differentiation trajectory, it is crucial to remove information-poor features prior to performing trajectory inference in order to obtain high quality embeddings, clustering assignments, or cellular orderings that are reproducible for both qualitative interpretation and downstream trajectory analysis.

**Quantitative assessment of feature selection method performance.** We next quantitatively examined how biological or technical challenges associated with single-cell RNA sequencing data may influence a feature selection method's ability to detect the particular genes that define cell types or cell type transitions. To do so, we systematically benchmarked the 12 described feature selection strategies on their capacity to preserve trajectories according to three sets of quantitative comparisons. Method performance was assessed by evaluating if selected genes from an approach were (1) differentially expressed within a cell type or along a lineage, (2) could be used to classify cell

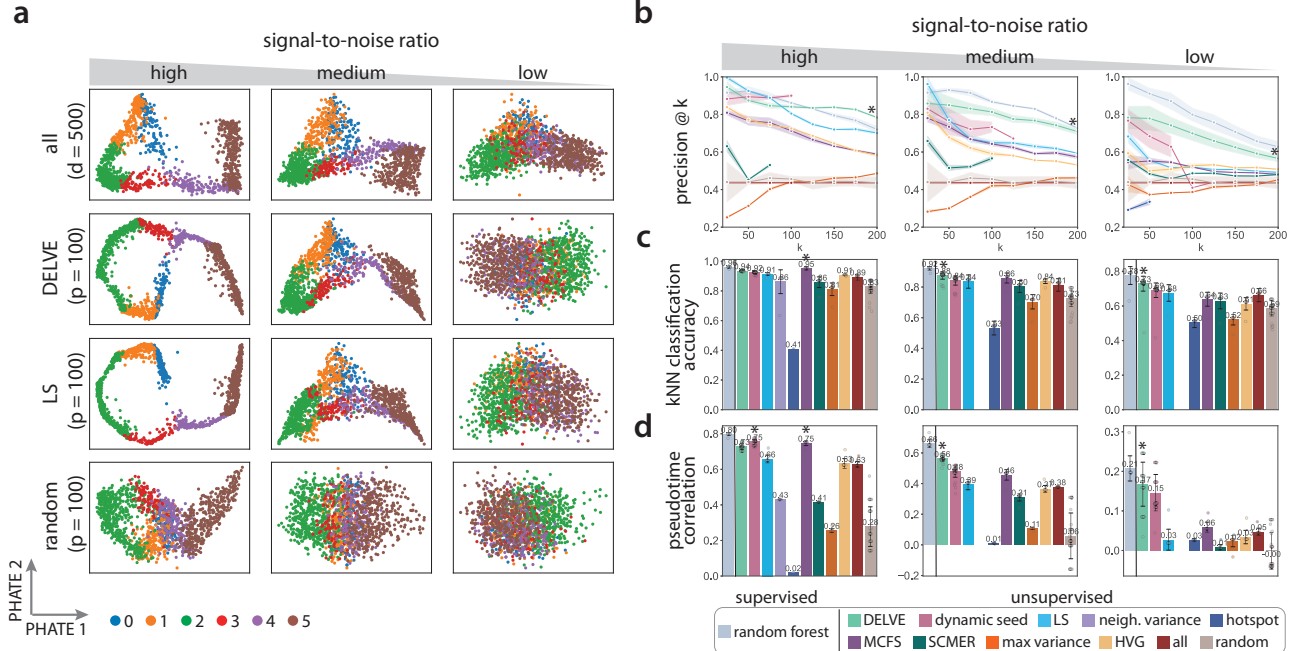

**Fig. 2 | Comparison of feature selection methods on preserving linear trajectories when subjected to a reduction in the signal-to-noise ratio. a** Example PHATE[78] visualizations of simulated linear trajectories using four feature selection approaches (all features, DELVE, Laplacian Score (LS)[55], and random selection) when subjected to a reduction in the signal-to-noise ratio (high, medium, low). Here, we simulated a reduction in the signal-to-noise ratio and stochastic gene expression by modifying the biological coefficient of variation (bcv) parameter within Splatter[75] that controls the mean-variance relationship between genes, where lowly expressed genes are more variable than highly expressed genes (high: bcv = 0.1, medium: bcv = 0.25, low: bcv = 0.5). *d* indicates the total number of genes (*d* = 500) and *p* indicates the number of selected genes following feature selection (*p* = 100). (**b**–**d**) Performance of twelve different feature selection

methods: random forest[49], DELVE, dynamic seed features, LS[55], neighborhood variance[29], hotspot[56], multi-cluster feature selection (MCFS)[58], single-cell manifold preserving feature selection (SCMER)[57], max variance, highly variable gene selection (HVG)[53], all features, random features. Following feature selection, trajectory preservation was quantitatively assessed according to several metrics: (**b**) the precision of differentially expressed genes at *k* selected genes, (**c**) *k*-NN classification accuracy, and (**d**) pseudotime correlation to the ground truth cell progression across 10 random trials. Error bands represent the standard deviation over *n* = 10 simulation datasets. Barplots show the mean ± the standard deviation over *n* = 10 simulation datasets. * indicates the method with the highest median score. For further details across other trajectory types and noise conditions, see Supplementary Figs. 1–9. Source data are provided in a Source Data file.

types, and (3) could accurately estimate individual cell progression through the cellular trajectory. Figure 2b–d shows feature selection method performance for simulated linear differentiation trajectories when subjected to the technical challenge of having a reduction in the signal-to-noise ratio.

First, we assessed the biological relevancy of selected genes, as well as the overall recovery of relevant genes as the signal-to-noise ratio decreased by computing a precision score. Precision@k is a metric that defines the proportion of selected genes (*k*) that are known to be differentially-expressed within a cell type or along a lineage (See Precision@k). Overall, we found that DELVE achieved the highest precision@k score between selected genes and the ground truth, validating that our approach was able to select genes that are differentially expressed and was the strongest in defining cell types and cell type transitions (See Fig. 2b). Importantly, DELVE's ability to recover informative genes was robust to the number of genes selected (*k*) and to the amount of noise present in the data. In contrast, variance-based, similarity-based, or subspace-learning approaches exhibited comparatively worse recovery of cell type and lineage-specific differentially expressed genes.

Given that a key application of single-cell profiling technologies is the ability to identify cell types or cell states that are predictive of sample disease status, responsiveness to drug therapy, or are correlated with patient clinical outcomes[68,79–82], we then assessed whether selected genes from a feature selection strategy can correctly classify cells according to cell type along the underlying cellular trajectory; this is a crucial and necessary step of trajectory analysis. Therefore, we trained a *k*-nearest neighbor (*k*-NN) classifier on the selected feature

set (see *k*-nearest neighbor classification) and compared the predictions to the ground truth cell type annotations by computing a cell type classification accuracy score. Across all simulated trajectories, we found that DELVE selected genes that often achieved the highest median *k*-NN classification accuracy score (high signal-to-noise ratio: 0.937, medium signal-to-noise ratio: 0.882, low signal-to-noise ratio: 0.734) and produced *k*-NN graphs that were more faithful to the underlying biology (See Fig. 2c). Moreover, we observed a few results that were consistent with the qualitative interpretations. First, removing irrelevant genes with DELVE, LS, or MCFS achieved higher *k*-NN classification accuracy scores (e.g., high signal-to-noise ratio; DELVE = 0.937, LS = 0.915, and MCFS = 0.955, respectively) than was achieved by retaining all genes (all = 0.900). Next, DELVE outperformed the Laplacian Score, suggesting that using a bottom-up framework and excluding noisy features prior to performing ranking and selection is crucial for recovering cell-type specific genes that would have been missed if the cell similarity graph was initially defined based on all genes. Lastly, when comparing the percent change in performance as the amount of noise corruption increased (e.g., high signal-to-noise ratio to medium signal-to-noise ratio) for linear trajectories, we found that DELVE often achieved the highest average classification accuracy score (0.905) and lowest percent decrease in performance (−6.398%), indicating that DELVE was the most robust unsupervised feature selection method to noise corruption (See Supplementary Fig. 10a). In contrast, the existing unsupervised similarity-based or subspace learning feature selection methods that achieved high to moderate average *k*-NN classification accuracy scores (e.g., MCFS = 0.905, LS = 0.874) had larger decreases in performance (e.g.,

MCFS = −9.673%, LS = −8.390%) as the amount of noise increased. This further highlights the limitations of current feature selection methods on identifying cell type-specific genes from noisy single-cell omics data.

Lastly, when undergoing dynamic biological processes such as differentiation, cells exhibit a continuum of cell states marked by linear and nonlinear changes in gene expression[83–85]. Therefore, we evaluated how well feature selection methods could identify genes that define complex differentiation trajectories and correctly order cells along the cellular trajectory in the presence of noise. To infer cellular trajectories and to estimate cell progression, we used the diffusion pseudotime algorithm[31] on the selected gene set from each feature selection strategy, as this approach has been shown previously[23] to perform reasonably well for inference of simple or branched trajectory types (See Trajectory inference and analysis). Method performance was then assessed by computing the Kendall rank correlation between estimated pseudotime and the ground truth cell progression. We found that DELVE approaches more accurately inferred cellular trajectories and achieved the highest median pseudotime correlation to the ground truth measurements, as compared to alternative methods or all features (See Fig. 2d). Furthermore, similar to the percent change in classification performance, we found that DELVE was the most robust unsupervised feature selection method in estimating cell progression, as it often achieved the highest average pseudotime correlation (0.645) and lowest percent decrease in performance (−22.761%) as the

amount of noise increased (See Supplementary Fig. 10b high to medium signal-to-noise ratio). In contrast, the alternative methods incorrectly estimated cellular progression and achieved lower average pseudotime correlation scores (e.g., MCFS = 0.602, LS = 0.526) and higher decreases in performance as the signal-to-noise ratio decreased (MCFS = −38.884%, LS = −40.208%).

We performed this same systematic evaluation across a range of trajectory types (e.g., linear, bifurcation, tree) and biological or technical challenges associated with single-cell data (See Supplementary Figs. 1–12). Figure 3 displays the overall ranked method performance of feature selection methods on preserving cellular trajectories when subjected to different sources of single-cell noise (pink: decreased signal-to-noise ratio, green: decreased library size, and blue: increased dropout). Ranked aggregate scores were computed by averaging results across all datasets within a condition; therefore, this metric quantifies how well a feature selection strategy can recover genes that define cell types or cell type transitions underlying a cellular trajectory when subjected to that biological or technical challenge (See Aggregate scores). Across all conditions, we found that DELVE often achieved an increased recovery of differentially expressed genes, higher cell type classification accuracy, higher correlation of estimated cell progression, and lower percent change in performance in noisy data. While feature selection method performance varied across biological or technical challenges, we found that the LS and MCFS performed reasonably well under low amounts of noise corruption and

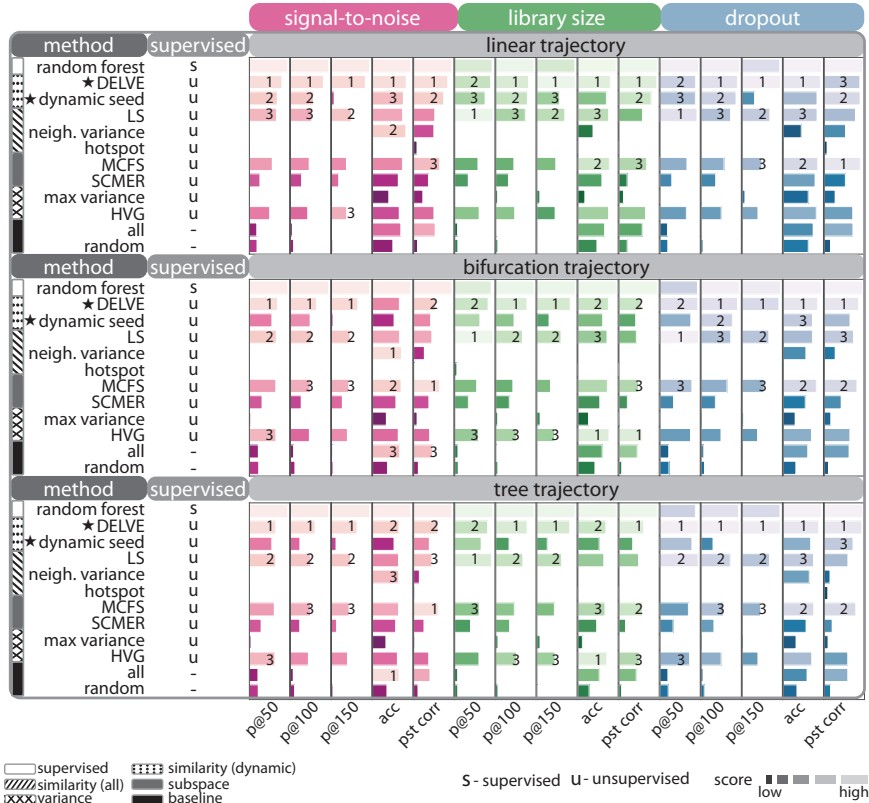

**Fig. 3 | DELVE outperforms existing feature selection methods on representing trajectories in the presence of single-cell RNA sequencing noise.** Feature selection methods were ranked by averaging their overall performance across datasets from different trajectory types (e.g., linear, bifurcation, tree) when subjected to noise corruption (e.g., decreased signal-to-noise ratio, decreased library size, and increased dropout). Several metrics were used to quantify trajectory preservation, including, precision of dynamically-expressed genes with 50 selected genes (p@50), precision at 100 selected genes (p@100), precision at 150 selected genes (p@150), k-NN classification accuracy of cell type labels (acc), and pseudotime correlation (pst). Here, higher-ranked methods are indicated by a longer

lighter bar, and the star illustrates our approach (DELVE) as well as the performance from dynamic seed features of step 1 of the algorithm. DELVE often achieves the highest precision of lineage-specific differentially expressed genes, the highest classification accuracy, and highest pseudotime correlation across noise conditions and trajectory types. Of note, random forest was included as a baseline representation to illustrate feature selection method performance when trained on ground truth cell type annotations; however, it was not ranked, as this study is focused on unsupervised feature selection method performance on trajectory preservation.

are often the second and third-ranked unsupervised methods. Altogether, this first simulation study with Splatter demonstrates that DELVE more accurately recapitulates cellular dynamics and can be used to effectively interrogate cell identity and lineage-specific gene expression dynamics from noisy single-cell data.

**SymSim single-cell RNA sequencing simulation study.** Given that single-cell RNA sequencing simulation software has the potential to generate parameters or distributions of counts that can fail to capture biologically-relevant data or the extent of technical limitations (e.g., efficiency of mRNA capture), we've additionally benchmarked feature selection methods on larger scale single-cell data using a secondary simulation approach SymSim[86]. This approach has been shown in a recent benchmarking study[87] to be amongst the top-ranking methods for reasonably simulating single-cell RNA sequencing data, as measured by the accuracy in estimating data properties (e.g., library size, TMM, mean expression, scaled variance, fractions of zeros, cell and gene correlation), as well as the ability to preserve biological signals (e.g., differentially expressed genes, differentially variable genes).

In this second study, we simulated five tree differentiation trajectories containing 10,000 cells, 20,000 genes, and 4 cell types by modifying the mean mRNA capture efficiency rate in SymSim (See SymSim simulation). Here, we aimed to evaluate how well feature selection methods could identify genes that define cellular trajectories when subjected to a reduction in the total mRNA count. Moreover, for each feature selection method, we additionally assessed the effect of selecting different numbers of features on trajectory preservation. Similar to the Splatter simulation study, overall, we found that similarity-based feature selection methods, DELVE, Laplacian score, and Hotspot outperformed the alternative feature selection methods and achieved higher cell type classification accuracy scores and higher correlations of estimated pseudotime to the ground truth cell progression (See Supplementary Fig. 13). Moreover, we observed that variance-based approaches (e.g., max variance, highly variable gene selection) required more features (e.g., 2000 genes) to obtain similar cell type classification accuracy and pseudotime correlation scores. In contrast, DELVE, Laplacian score, and Hotspot achieved higher scores with smaller representative subsets of features (e.g., 100 genes) and were more robust to the number of selected genes (See Supplementary Fig. 13). Furthermore, neighborhood variance and SCMER identified the smallest number of genes (e.g., 50 genes), and they were often biologically predictive. Overall, these results suggest that unsupervised similarity-based feature selection methods outperform variance-based approaches in preserving cellular trajectories when evaluated on simulated single-cell RNA sequencing data.

## Revealing molecular trajectories of proliferation and cell cycle arrest

Recent advances in spatial single-cell profiling technologies[8–11,88–92] have enabled the simultaneous measurement of transcriptomic or proteomic signatures of cells, while also retaining additional imaging or array-derived features that describe the spatial positioning or morphological properties of cells. These spatial single-cell modalities have provided fundamental insights into mammalian organogenesis[92,93] and complex immune responses linked to disease progression[21,94]. By leveraging imaging data to define cell-to-cell similarity, DELVE can identify smoothly varying spatial features that are strongly associated with cellular progression, such as changes in cell morphology or protein localization, while excluding information-poor, noisy, or irrelevant imaging-derived features that can obfuscate the underlying cellular trajectory.

To demonstrate this, we applied DELVE to an integrated live cell imaging and protein iterative indirect immunofluorescence imaging (4i) dataset consisting of 2759 human retinal pigmented epithelial cells (RPE) undergoing the cell cycle (See RPE analysis). In a recent study[17],

we performed time-lapse imaging on an asynchronous population of non-transformed RPE cells expressing a PCNA-mTurquoise2 reporter to record the cell cycle phase (G0/G1, S, G2, M) and age (time since last mitosis) of each cell. We then fixed the cells and profiled them with 4i to obtain measurements of 48 core cell cycle effectors. The resultant dataset consisted of 241 imaging-derived features describing the expression and localization of different protein markers (e.g., nucleus, cytoplasm, perinuclear region—denoted as ring), as well as morphological measurements from the images (e.g., size and shape of the nucleus). Given that time-lapse imaging was performed prior to cell fixation, this dataset provides the unique opportunity to rigorously evaluate feature selection methods on a real biological system (cell cycle) with technical challenges (e.g., many features with low signal-to-noise ratio, autofluorescence, sample degradation). Moreover, it's important to note that although all profiled proteins in this study were indeed cell-cycle specific, many were not expressed within all regions of the cell, and many of the extracted imaging-derived features were not relevant or biologically predictive of cell cycle progression[17]. Thus, the goal of this evaluation is to assess if feature selection methods can be used to identify proteomic-imaging derived features that are strongly associated with cell cycle progression from a feature list that contains all extracted imaging measurements (noisy or otherwise).

We first tested whether DELVE can identify a set of dynamically-expressed cell cycle-specific imaging-derived features to construct an approximate cellular trajectory graph for feature selection. Overall, we found that DELVE successfully identified dynamically-expressed seed features ($p = 13$ out of 241 total imaging-derived features) that are known to be associated with cell cycle proliferation (e.g., increase in DNA content and area of the nucleus) and captured key mechanisms previously shown to drive cell cycle progression (Fig. 4a right), including molecular events that regulate the G1/S and G2/M transitions. For example, the G1/S transition is governed by the phosphorylation of RB by cyclin:CDK complexes (e.g., cyclinA/CDK2 and cyclinE/CDK2), which control the expression of E2F transcription factors that regulate S phase genes[95]. We also observed an increase in expression of Skp2, which reduces p27-mediated inhibition of E2F1 target genes[96,97]. In addition, our approach identified S phase events that are known to be associated with DNA replication, including an accumulation of PCNA foci at sites of active replication[98] and a DNA damage marker, pH2AX, which becomes phosphorylated in response to double-stranded DNA breaks in areas of stalled replication[99,100]. Lastly, we observed an increase in expression of cyclin B localized to different regions of the cell, which is a primary regulator of G2/M transition alongside CDK1[101,102]. Of note, phosphorylation of RB also controls cell cycle re-entry and is an important biomarker that is often used for distinguishing proliferating from arrested cells[103,104]. Furthermore, by ordering the average pairwise change in expression of features across ground truth phase annotations, we observed that DELVE dynamically-expressed seed features exhibited non-random patterns of expression variation that gradually increased throughout the canonical phases of the cell cycle (Fig. 4a), and were amongst the top-ranked features that were biologically predictive of cell cycle phase and age measurements using a random forest classification and regression framework, respectively (See Random forest, Fig. 4a right, Supplementary Fig. 16). Collectively, these results illustrate that the dynamic feature module identified by DELVE represents a minimum cell cycle feature set (Fig. 4b dynamic seed) that precisely distinguishes individual cells according to their cell cycle progression status and can be used to construct an approximate cellular trajectory for ranking feature importance.

We then comprehensively evaluated feature selection methods on their ability to retain imaging-derived features that define cell cycle phases and resolve proliferation and arrest cell cycle trajectories. We reasoned that cells in similar stages of the cell cycle (as defined by the cell cycle reporter) should have similar cell cycle signatures (4i

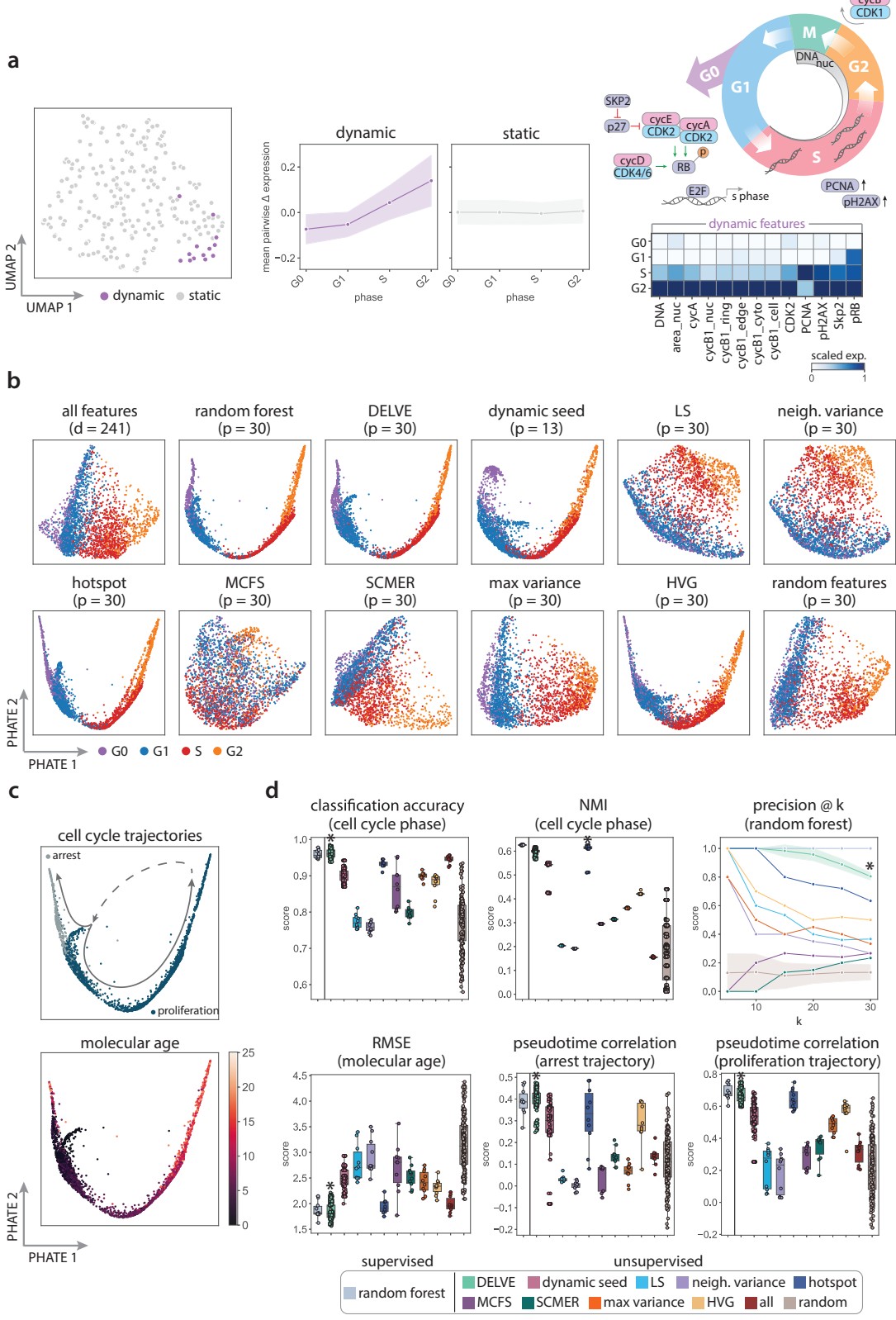

features) and should be located near one another in a low-dimensional projection. Figure 4b shows the PHATE embeddings from each feature selection strategy. Using the DELVE feature set, we obtained a continuous PHATE trajectory structure that successfully captured the smooth progression of cells through the canonical phases of the cell cycle, where cells were tightly grouped together according to ground truth cell cycle phase annotations (Fig. 4b). Moreover, we observed

that the two DELVE approaches (i.e., DELVE and dynamic seed), in addition to hotspot and HVG selection, produced qualitatively similar denoised lower-dimensional visualizations comparable to the supervised random forest approach that was trained on ground truth cell cycle phase annotations. In contrast, similarity-based approaches such as LS and neighborhood variance, which define a cell similarity graph according to all features, showed more diffuse presentations of cell

**Fig. 4 | DELVE recovered signatures of proliferation and arrest in noisy protein immunofluorescence imaging data. a** DELVE identified one dynamic module consisting of 13 seed features that represented a minimum cell cycle. (a left) UMAP visualization of image-derived features where each point indicates a dynamic or static feature identified by the model. (a middle) The average pairwise change in expression of features within DELVE modules ordered across ground truth cell cycle phase annotations. (a right top) Simplified signaling schematic of the cell cycle highlighting the role of DELVE dynamic seed features within cell cycle progression. (a right bottom) Heatmap of the standardized average expression of dynamic seed features across cell cycle phases. **b** Feature selection was performed to select the top ($p = 30$) ranked features from the original ($d = 241$) feature set according to a feature selection strategy. PHATE visualizations illustrating the overall quality of low-dimensional cell cycle trajectories following feature selection. Cells were labeled according to cell cycle phase annotations from time-lapse imaging. **c** PHATE visualizations following DELVE feature selection, where cells were labeled according to cell cycle trajectory (top) or age measurements (bottom).

**d** Performance of feature selection methods on representing the cell cycle according to several metrics, including cell phase classification accuracy, normalized mutual information (NMI), precision of phase-specific features, root mean squared error (RMSE) between predicted and ground truth age, and the correlation between estimated pseudotime to the ground truth age measurements following trajectory inference. All error bands represent the standard deviation. All boxplots show the median (middle line), the interquartile range (upper and lower bounds of the box), and the minimum and maximum of the distributions (whiskers) over $n = 10$ random splits, seeds, or root cells. DELVE, dynamic seed, and random feature selection were run over $n = 20$ random trials to show reproducibility of the approach. * indicates the method with the highest median score. DELVE achieved the highest classification accuracy, highest p@k score, high NMI clustering score, lowest RMSE, and highest correlation of estimated pseudotime to the ground truth age indicating robust prediction of cell cycle transitions. Source data are provided in a Source Data file.

states. Variance-based (max variance) or subspace-learning approaches (SCMER, MCFS) produced qualitatively similar embeddings to that produced using all features.

To quantitatively assess if selected features from a feature selection strategy were biologically predictive of cell cycle phases, we performed three complementary analyses. We first focused on the task of cell state classification, where our goal was to learn the ground truth cell cycle phase annotations from the selected feature set. To do so, we trained a support vector machine (SVM) classifier and compared the accuracy of predictions to their ground truth phase annotations (See Support Vector Machine). We performed nested tenfold cross-validation to obtain a distribution of predictions for each method. Overall, we found that DELVE achieved the highest median classification accuracy (DELVE = 0.960) obtaining a similar performance to the random forest classifier trained on cell cycle phase annotations (random forest = 0.957), and outperforming existing unsupervised approaches (e.g., hotspot = 0.935, max variance = 0.902, HVG = 0.889, MCFS = 0.870, SCMER = 0.797, LS = 0.770), as well as all features (0.946), suggesting that selected features with DELVE were more biologically predictive of cell cycle phases (Fig. 4d). We next aimed to assess how well a feature selection method could identify and rank cell cycle phase-specific features according to their representative power. To test this, we trained a random forest classifier on the ground truth phase annotations using nested 10-fold cross validation (See Random forest). We then compared the average ranked feature importance scores from the random forest to the selected features from a feature selection strategy using the precision@k metric. Strikingly, we found that DELVE achieved the highest median precision@k score (DELVE $p$@30 = 0.800) and appropriately ranked features according to their discriminative power of cell cycle phases despite being a completely unsupervised approach (Fig. 4d). This was followed by hotspot with a precision@k score of (hotspot $p$@30 = 0.633) and highly variable gene selection (HVG $p$@30 = 0.500). In contrast, the Laplacian Score and max variance obtained low precision scores ($p$@30 = 0.367 and 0.333 respectively), whereas neighborhood variance and subspace-learning feature selection methods MCFS and SCMER were unable to identify cell cycle phase-specific features from noisy 4i data and exhibited precision scores near random ($p$@30 = 0.267, 0.267, and 0.233, respectively). Lastly, we assessed if selected image-derived features could be used for downstream analysis tasks like unsupervised cell population discovery. To do so, we clustered cells using the KMeans++ algorithm[105] on the selected feature set and compared the predicted labels to the ground truth annotations using a normalized mutual information (NMI) score over 25 random initializations (See Unsupervised clustering). We found that hotspot, DELVE, and dynamic seed features were better able to cluster cells according to cell cycle phases and achieved considerably higher median NMI scores (0.615, 0.599, 0.543, respectively), as compared to retaining all features (0.155)

(Fig. 4d). Moreover, we found that clustering performance was similar to that of the random forest trained on cell cycle phase annotations (0.626). In contrast, variance-based approaches achieved moderate NMI clustering scores (HVG: 0.421, max variance: 0.361) and alternative similarity-based and subspace learning approaches obtained low median NMI scores (-0.2) and were unable to cluster cells into biologically-cohesive cell populations. Of note, many trajectory inference methods use clusters when fitting trajectory models[27,28,30,106], thus accurate cell-to-cluster assignments following feature selection is crucial for both cell type annotation and discovery, as well as for accurate downstream trajectory analysis interpretation. Collectively, these results highlight that feature selection with DELVE identifies imaging-derived features from noisy protein immunofluorescence imaging data that are more biologically predictive of cell cycle phases.

We then focused on the much harder task of predicting an individual cell's progression through the cell cycle. A central challenge in trajectory inference is the destructive nature of single-cell technologies, where only a static snapshot of cell states is profiled. To move toward a quantitative evaluation of cell cycle trajectory reconstruction following feature selection, we leveraged the ground truth age measurements determined from time-lapse imaging of the RPE-PCNA reporter cell line. We first evaluated whether selected features could be used to accurately predict cell cycle age by training an SVM regression framework using nested tenfold cross-validation (See Support Vector Machine). Method performance was subsequently assessed by computing the root mean squared error (RMSE) between the predicted and the ground truth age measurements. Overall, we found that DELVE achieved the lowest median RMSE (1.806 h), outperforming both supervised (random forest = 1.815 h) and unsupervised approaches (e.g., second-best performer hotspot = 1.911 h) suggesting that selected features more accurately estimate the time following mitosis (Fig. 4c). Crucially, this highlights DELVE's ability to learn new biologically-relevant features that might be missed when performing a supervised or unsupervised approach. Lastly, we assessed whether selected imaging features could be used to accurately infer proliferation and arrest cell cycle trajectories using common trajectory inference approaches (Fig. 4d). Briefly, we constructed predicted cell cycle trajectories using the diffusion pseudotime algorithm[31] under each feature selection strategy (See Trajectory inference and analysis). Cells were separated into proliferation or arrest lineages according to their average expression of pRB, and cellular progression was estimated using ten random root cells that had the youngest age. Feature selection method performance on trajectory inference was then quantitatively assessed by computing the Kendall rank correlation between estimated pseudotime and the ground truth age measurements. We found that DELVE achieved the highest median correlation of estimated pseudotime to the ground truth age measurements (proliferation: 0.656, arrest: 0.405) as compared to alternative

methods (second best performer hotspot; proliferation: 0.632, arrest: 0.333) or all features (proliferation: 0.330, arrest: 0.135), indicating that our approach was better able to resolve both proliferation and cell cycle arrest trajectories where other approaches failed (Fig. 4d). Of note, DELVE was robust to the choice in hyperparameters and obtained reproducible results across a range of hyperparameter choices (See Guidelines on parameter selection, Supplementary Fig. 15).

To further investigate the differences in performance between similarity-based feature selection methods DELVE, Hotspot, and the Laplacian score on 4i data in more detail, we evaluated the ability for feature selection methods to recover cell cycle-defining features from immunofluorescence imaging data with different amounts of noisy variables (See Supplementary Fig. 17). To do so, we generated multiple RPE immunofluorescence imaging datasets, where each dataset was initialized with the same set of ground truth phase-specific features (i.e., the top 30 predictive features of phase by training a random forest classifier on phase labels from time-lapse imaging). We then added a fixed amount of noisy variables (ranging from $j = 100-500$) to each dataset by randomly sampling features from the experimental RPE dataset with replacement. For each dataset and feature selection method, we selected the top 30 ranked features and compared the performance of DELVE, Hostpot, and the Laplacian score on various trajectory tasks. Overall, DELVE achieved the highest recovery of phase-specific features, cell population recovery (NMI between cluster labels and ground truth phase annotations), and trajectory recovery (highest correlation between estimated pseudotime and ground truth age annotations). In contrast, as the amount of noisy variables increased, Hotspot and the Laplacian score identified less phase-specific features and more noisy features, which resulted in worse cell population and trajectory recovery. These results demonstrate the necessity of using a bottom-up approach when performing feature selection from noisy proteomic imaging data and showcases the robustness of DELVE on creating a representation of the data that is faithful to underlying cellular trajectory structure.

As a secondary validation, we applied DELVE to nine pancreatic adenocarcinoma (PDAC) cell lines (e.g., BxPC3, CFPAC, MiaPaCa, HPAC, Pa01C, Pa02C, PANC1, UM53) profiled with 4i (See PDAC analysis) and performed a similar evaluation of cell cycle phase and phase transition preservation (See Supplementary Figs. 19–27). Across all cell lines and metrics, we found that DELVE approaches and hotspot considerably outperformed alternative methods on recovering the cell cycle from noisy 4i data and often achieved the highest classification accuracy scores, clustering scores, and the highest correlation of cellular progression along proliferative and arrested cell cycle trajectories (See Supplementary Figs. 18, 28). Notably, DELVE was particularly useful in resolving cell cycle trajectories from the PDAC cell lines that had numerous imaging measurements with low signal-to-noise ratio (e.g., CFPAC, MiaPaCa, PANC1, and UM53), whereas the alternative strategies were unable to resolve cell cycle phases and achieved scores near random (See Supplementary Figs. 20, 22, 26, 27).

## Identifying molecular drivers of CD8+ T cell effector and memory formation

To demonstrate the utility of our approach in a complex differentiation setting consisting of heterogeneous cell subtypes and shared and distinct molecular pathways, we applied DELVE to a single-cell RNA sequencing time series dataset consisting of 29,893 mice splenic CD8+ T cells responding to acute viral infection[107]. Here, CD8+ T cells were profiled over 12-time points following infection with the Armstrong strain of lymphocytic choriomeningitis virus (LCMV): Naive, d3-, d4-, d5-, d6-, d7-, d10-, d14-, d21-, d32-, d60-, and d90- post-infection (See CD8+ T cell differentiation analysis). During an immune response to acute viral infection, naive CD8+ T cells undergo a rapid activation and proliferation phase, giving rise to effector cells that can serve in a

cytotoxic role to mediate immediate host defense, followed by a contraction phase giving rise to self-renewing memory cells that provide long-lasting protection and are maintained by antigen-dependent homeostatic proliferation[108–110]. Despite numerous studies detailing the molecular mechanisms of CD8+ T cell effector and memory fate specification, the molecular mechanisms driving activation, fate commitment, or T cell dysfunction continue to remain unclear due to the complex intra- and inter-temporal heterogeneity of the CD8+ T cell response during infection. Therefore, we applied DELVE to the CD8+ T cell dataset to resolve the differentiation trajectory and investigate transcriptional changes that are involved in effector and memory formation during acute viral infection with LCMV.

Following unsupervised seed selection, we found that DELVE successfully identified three gene modules constituting core regulatory complexes involved in CD8+ T cell viral response and had dynamic expression patterns that varied across experimental time following viral infection (Fig. 5a–c). Namely, dynamic module 0 contained genes involved in early activation and interferon response (e.g., Ly6a, Bst2, Ifi27l2a)[111,112], and proliferation (e.g., Cenpa, Cenpf, Ccnb2, Ube2c, Top2a, Tubb4b, Birc5, Cks2, Cks1b, Nusap1, Hmgb2, Rrm2, H2afx, Pclaf, Stmn1, Lbr, Smc2, Cdc20, Hmgn2, Cbx3, Ube2s, Mki67, Cdk1, Ptma)[113]. Dynamic module 1 contained genes involved in effector formation, including interferon-γ cytotoxic molecules, such as perforin/granzyme pathway (e.g., Gzma, Gzmk), integrins (e.g., Itga4, Itgax, Itgb1), killer cell lectin-like receptor family (e.g., Klrg1, Klrd1, Klrk1, Klrc1, klrc2), cytokine and chemokine receptors (e.g., Il18r1, Cxcr3, Cxcr6, Ccr2), and canonical transcription factors involved in terminal effector formation (e.g., Id2, Klf2, Klf3, Zeb2)[114–117]. Lastly, dynamic module 2 contained genes involved in long-term memory formation (e.g., Bcl2, Il7r, Ltb, Tcf7, Btg1, Btg2)[118–121]. To quantitatively examine if genes within a dynamic module were meaningfully associated with one another, or had experimental evidence of co-regulation, we constructed gene association networks using experimentally-derived association scores from the STRING database[122]. Here, a permutation test was performed to assess the statistical significance of the observed experimental association amongst genes within a DELVE module as compared to random gene assignment (See Protein-protein interaction networks). Notably, across all three dynamic modules, DELVE identified groups of genes that had statistically significant experimental evidence of co-regulation ($p$-value = 0.001), where DELVE networks had a larger average degree of experimentally-derived edges than the null distribution (Fig. 5b: dynamic modules). Degree centrality is a simple measurement of the number of edges (e.g., experimentally derived associations between genes) connected to a node (e.g., gene); therefore, in this context, networks with a high average degree may contain complexes of genes that are essential for regulating a biological process. In contrast, genes identified by DELVE that exhibited random or noisy patterns of expression variation (static module) had little to no evidence of co-regulation ($p$ value = 1.0) and achieved a much lower average degree than networks defined by random gene assignment (Fig. 5b).

Next, we evaluated if the dynamically-expressed genes that had experimental evidence of co-regulation from DELVE could be used to improve the identification of molecular pathways associated with long-term CD8+ T cell memory formation following trajectory inference, as compared to the standard approach of highly variable gene selection. To do so, we reconstructed the CD8+ T cell differentiation trajectory using either the diffusion pseudotime algorithm[31] or Slingshot[28] on the top 500 ranked genes from a feature selection strategy (Fig. 5d–e, Supplementary Fig. 29a–b). Here, we considered both similarity-based feature selection approaches (e.g., DELVE, Laplacian score, and Hotspot), as well as highly variable gene selection. We then performed a regression analysis for each gene along estimated pseudotime using a negative binomial generalized additive model (GAM). Genes were considered to be differentially expressed along the memory lineage if

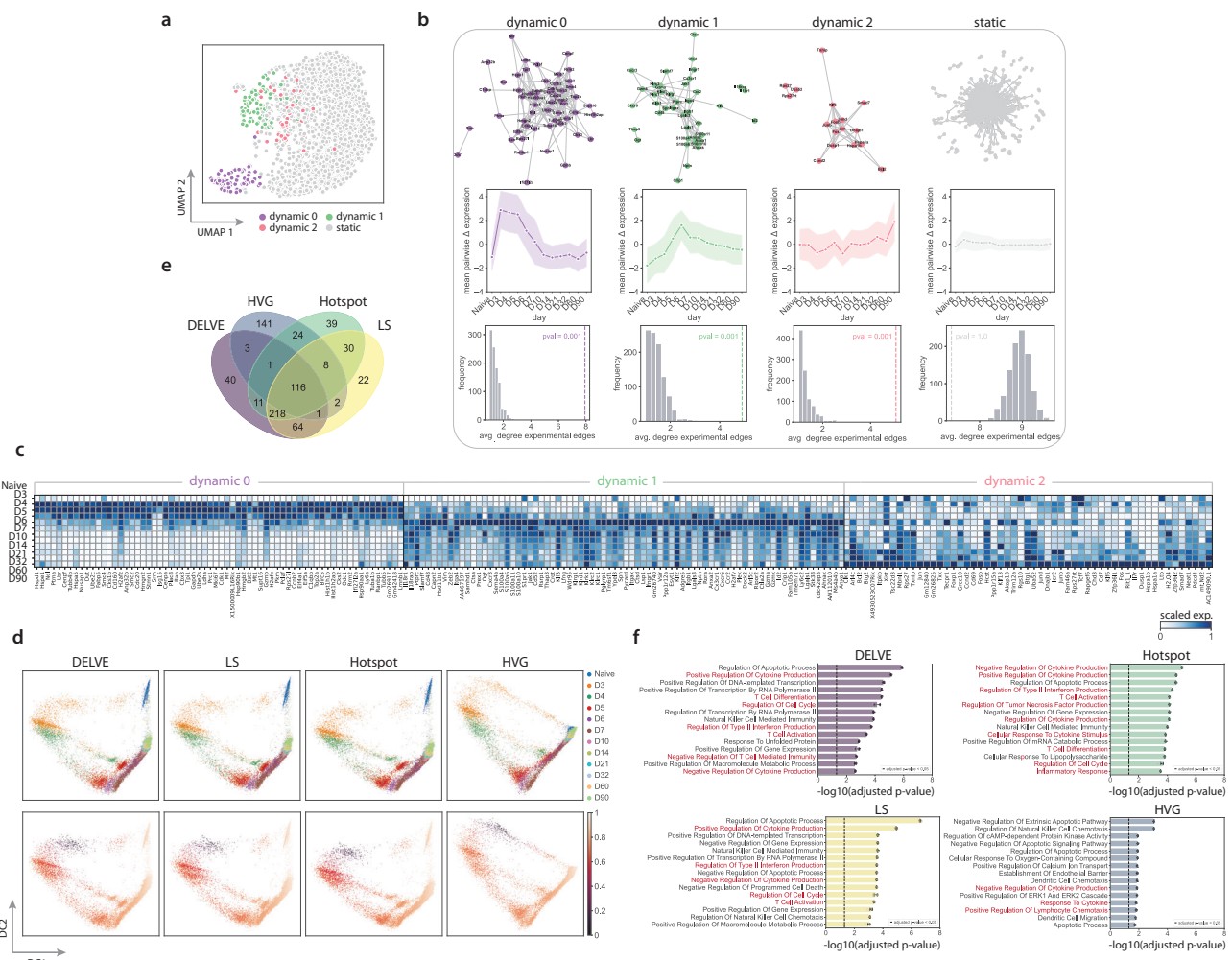

**Fig. 5 | DELVE identified molecular drivers of CD8+ T cell effector and memory formation. a** DELVE identified three dynamic modules representing cell cycle and early activation (dynamic 0), effector formation and cytokine signaling (dynamic 1), and long-term memory formation (dynamic 2) during CD8+ T cell differentiation response to viral infection with lymphocytic choriomeningitis virus (LCMV). UMAP visualization of ($d$ = 2000) genes where each point indicates a dynamic or static gene identified by the model. **b** A permutation test was performed using experimentally-derived association scores from the STRING interaction database[122] to assess whether genes within DELVE dynamic modules had experimental evidence of co-regulation as compared to random assignment. (**b** top) STRING association networks, where nodes represent genes from a DELVE module and edges represent experimental evidence of association. (**b** middle) Average pairwise change in expression amongst genes within a module ordered by time following infection. Error bands represent the standard deviation. (**b** bottom) Histograms showing the distribution of the average degree of experimentally-derived edges of gene networks from $R$ = 1000 random permutations. The dotted line indicates the observed average degree from genes within a DELVE module. $p$ values were computed using a one-sided permutation test. **c** Heatmap visualization of the standardized average expression of dynamically-expressed genes identified by DELVE ordered across time following infection. **d** Diffusion map visualizations of the CD8+ T cell memory lineage for four feature selection strategies (DELVE, Laplacian score, Hotspot, highly variable gene (HVG) selection). Cells were colored according to (top) time following infection (bottom) estimated pseudotime using the diffusion pseudotime algorithm[31]. **e** Genes were regressed along estimated pseudotime using a generalized additive model to determine lineage-specific significant genes. The venn diagram illustrates the quantification and overlap of memory lineage-specific genes across feature selection strategies. **f** The barplots show the mean ± standard deviation for the top 15 gene set enrichment terms associated with the temporally-expressed gene lists specific to each feature selection strategy for $n$ = 10 random root cells. $p$-values were computed using a Fisher exact test and adjusted with Benjamini Hochberg multiple testing hypothesis correction. Source data are provided in a Source Data file.

they had a $q$ value < 0.05 following Benjamini-Hochberg false discovery rate correction[123] (See Trajectory inference and analysis). Overall, we found that ordering cells according to similarities in selected gene expression using similarity-based feature selection methods such as DELVE, Laplacian score, and Hotspot were more reflective of long-term memory formation and achieved an increased recovery of memory lineage-specific genes, as directly compared to the standard approach of highly variable gene selection (Fig. 5d–f, Supplementary Fig. 29).

To determine the biological relevance of these memory lineage-specific genes, we performed gene set enrichment analysis on the temporally-expressed genes specific to each feature selection strategy

using EnrichR[124]. Here, DELVE, Laplacian score, and Hotspot obtained higher significance scores and identified more terms involved in immune regulation and memory CD8+ T cell formation, including, T cell differentiation, T cell activation, regulation of cell cycle, regulation of cytokine production, regulation of type II interferon production, negative regulation if T cell-mediated immunity (see DELVE as compared to HVG in Fig. 5f, Supplementary Fig. 29c).

### Characterizing human embryonic stem cell differentiation into the definitive endoderm

During cellular differentiation, cells exhibit a continuum of cell states with fate transitions marked by external stimuli, cell-cell interactions,

and linear and nonlinear gene expression[83–85]. Given the recent advances in computational methods for inferring gene expression dynamics[125,126], we next sought to understand whether DELVE can be used to identify known dynamic regulators or transcription factors driving cell fate specification. Towards this objective, we applied DELVE to a multiplexed single-cell RNA sequencing dataset consisting of 5397 human embryonic stem cells differentiating into the defintive endoderm, an early lung precursor cell type.

Although single-cell RNA sequencing is a powerful method for capturing a variety of nuanced cell states, characterizing differentiation trajectories typically requires the use of multiple samples, which can be fraught with confounding variables like sample-specific batch effects[127]. To overcome this limitation and more rigorously investigate the transcriptional changes that occur during directed definitive endoderm differentiation, we adapted a multiplexed single-cell RNA sequencing approach[128] to profile the differentiation trajectory along three key stages, including, pluripotency, primitive streak, and definitive endoderm (DE). Briefly, H9 hESCs were differentiated into the definitive endoderm by inducing the TGFβ and WNT signaling pathways with small molecules Activin A and CHIR99021 over a two-day time course[129] (See Fig. 6a). Next, cell surface proteins were chemically labeled with unique timepoint-specific oligo barcodes using click chemistry. Lastly, labeled cells were pooled prior to cDNA library preparation and single-cell RNA sequencing using the 10X Genomics Chromium platform (See DE differentiation analysis).

Following unsupervised seed selection, we found that DELVE successfully identified two dynamic gene modules involved in the loss of pluripotency and formation of the primitive streak (dynamic 0), as well as the specification of the definitive endoderm (dynamic 1) (Fig. 6b–d). More specifically, dynamic module 0 contained genes involved in the regulation of pluripotency (POU5F1, SOX2, UTF1, LECT1, EPCAM, UCHL1, FGF2, ESRP1, DIAPH2)[130–132], cell cycle (CCND1)[133], and primitive streak formation (CDH1)[134]. In contrast, dynamic module 1 contained genes involved in early mesendoderm (e.g., MIXL1, GSC, MESP1, EOMES, LHX1, FST)[135–137], mesoderm (e.g., BMP4, IRX3, HAND1)[138,139], and definitive endoderm formation (e.g., GATA6, SOX17, LEFTY1, LEFTY2, CER1, RHOC, CYP26A1, APLNR, SFRP1)[140–142]. Similar to the previous analysis, we then quantitatively examined if the genes with each dynamic module had experimental evidence of co-regulation by (1) constructing gene association networks using the experimentally-derived association scores from the STRING database[122] and (2) performing a permutation test to assess the statistical significance of observed experimental associations. We found that DELVE identified groups of genes with significant experimental evidence of association ($p$ value = 0.033, $p$ value = 0.001), where DELVE networks had larger average degrees of experimentally-derived edges as compared to the null distribution (Fig. 6c: dynamic modules). Moreover, noisy genes identified by DELVE had little to no experimental evidence of co-regulation ($p$ value = 0.997) (Fig. 6c: static module). Overall, these results highlight DELVE's sensitivity in capturing a core set of gene regulatory complexes involved in pluripotency and cell fate specification.

We next sought to quantify continuous gene expression dynamics associated with definitive endoderm fate commitment. To do so, we performed trajectory inference using either the diffusion pseudotime algorithm[31] or Slingshot[28] on the top 500 ranked genes from each feature selection strategy. Here, we compared similarity-based feature selection methods (DELVE, Laplacian score, Hospot) to variance-based selection (HVG), as well as the top-ranked likelihood genes from RNA velocity. By modeling the conversion between unspliced pre mRNA and spliced mature mRNA molecules in a transcription-based kinetic model, RNA velocity[125,126] has been used to predict future gene expression changes in individual cells. Here, the dynamical model of RNA velocity computes a likelihood for each gene, where genes are then ranked according to their goodness of fit. This is used to identify genes that exhibit splicing dynamics and might be candidate regulators of the underlying dynamic process[126]. Following feature selection, we performed a regression analysis for each gene along estimated pseudotime using a negative binomial generalized additive model, where genes were considered to be differentially expressed if they had a $q$ value < 0.05 following Benjamini-Hochberg false discovery rate correction[123] (See Trajectory inference and analysis).

Overall, we found that similarity-based feature selection methods (DELVE, Laplacian score, Hotspot) identified more genes involved in definitive endoderm specification, as directly compared to both highly variable gene selection and RNA velocity ranked likelihood genes (See Supplementary Figs. 30–31). When comparing gene set enrichment analysis results for the temporally-expressed genes specific to DELVE, HVG, or RNA velocity using EnrichR[124], we found that DELVE obtained much higher significance scores and identified more gastrulation and definitive endoderm pathway-related terms, including endoderm formation, differentiation, and development; regulation of cell migration; negative regulation of canonical WNT signaling; gastrulation; anterior/posterior axis specification; and regulation of cell population proliferation (See DELVE as compared to HVG and RNA velocity in Fig. 6e–f). Moreover, of those temporally expressed genes, DELVE identified ~30% more lineage-specific transcription factors (See Fig. 6g) than HVG selection and RNA velocity. Strikingly, when comparing the transcription factors specific to each feature selection approach using the AnimalTFDB transcription factor database[143], we found that RNA velocity often failed to appropriately model and identify key transcription factors driving definitive endoderm differentiation, as they exhibited more switch-like or transient kinetic behavior (See Fig. 6h). In contrast, DELVE was able to successfully identify these dynamic transcription factors involved the (1) core pluripotency network (SOX2, POU5F1)[130,144,145], (2) organization and formation of the primitive streak and mesendoderm (CDX1, CDX2, EOMES, GSC, OTX2, MESP1, MESP2, LHX1)[63,137,146–150], and (3) known regulators involved in DE cell fate specification (SOX17, FOXA2)[151–153] (See Fig. 6h).

## Discussion

Computational trajectory inference methods have transformed our ability to study the continuum of cellular states associated with dynamic phenotypes; however, current approaches for reconstructing cellular trajectories can be hindered by biological or technical noise inherent to single-cell data[45,46]. To mitigate the effect of unwanted sources of variation confounding trajectory inference, we designed a bottom-up unsupervised feature selection method that ranks and selects features that best approximate cell state transitions from dynamic feature modules that constitute core regulatory complexes. The key innovation of this work is the ability to parse temporally co-expressed features from noisy information-poor features prior to performing feature selection; in doing so, DELVE constructs cell similarity graphs that are more reflective of cell state progression for ranking feature importance.

In this study, we benchmarked twelve feature selection methods[29,49,53,55–58] on their ability to identify biologically relevant features for trajectory analysis from single-cell RNA sequencing data and proteomic imaging data. In the context of simulated single-cell RNA sequencing data where the ground truth was known, we found that similarity-based feature selection methods (e.g., DELVE, Laplacian score, and Hotspot) achieved the highest recovery of differentially expressed genes within a cell type or along a cellular lineage, highest cell type classification accuracy, and most accurately estimated individual cell progression across a variety of trajectory topologies and biological or technical challenges. Furthermore, through a series or qualitative and quantitative comparisons, we illustrated how noise (e.g., stochasticity, sparsity, low library size) and information-poor features can create spurious similarities amongst cells and considerably impact the performance of existing subspace learning-based

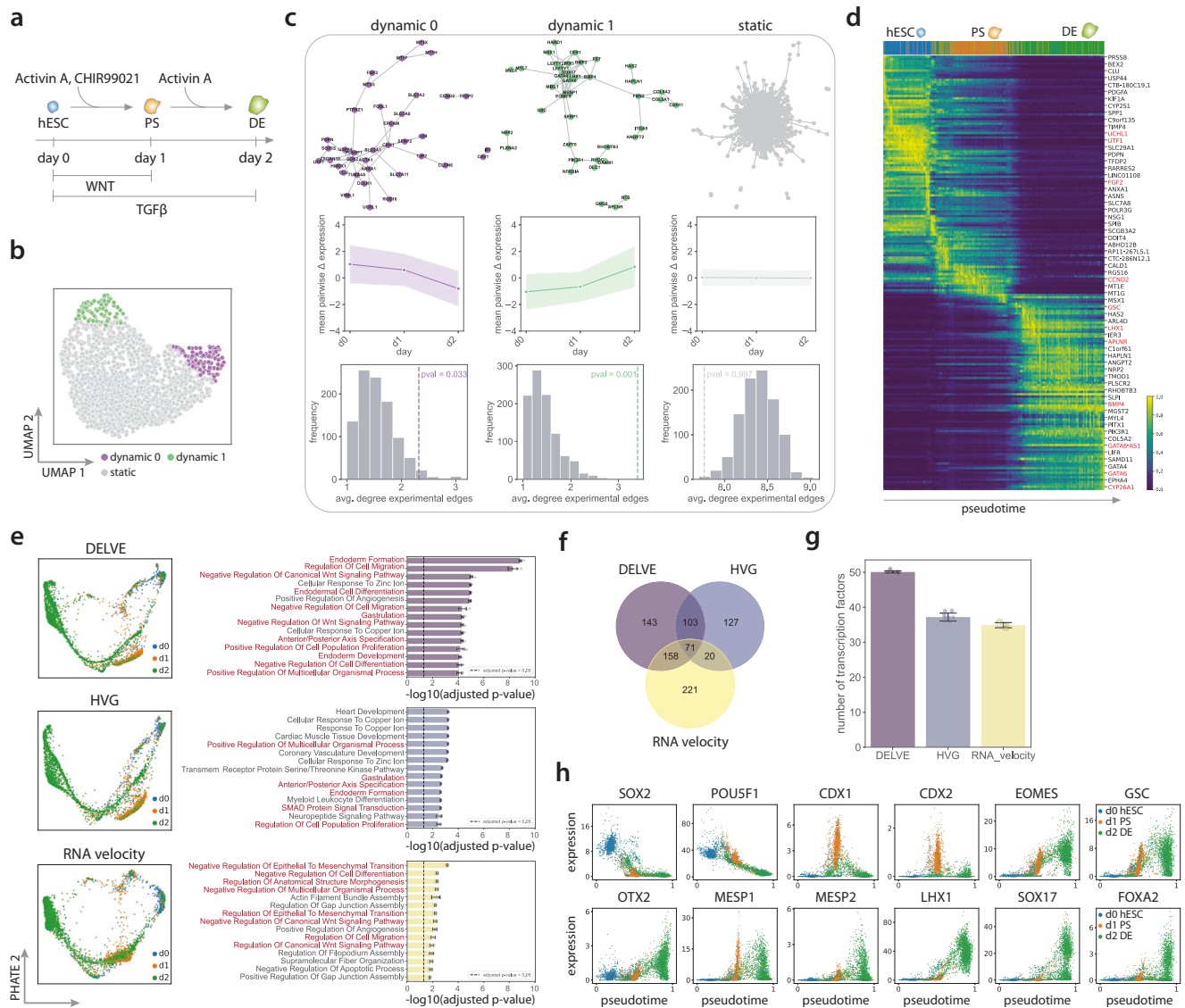

**Fig. 6 | Characterizing human embryonic stem cell differentiation into the definitive endoderm. a** Schematic of the definitive endoderm differentiation single-cell RNA sequencing design. Human embryonic stem cells (hESC) were differentiated into the definitive endoderm (DE) over a two day time course. **b** DELVE identified two dynamic modules representing the pluripotency to primitive streak (PS) transition (dynamic 0) and DE formation (dynamic 1). UMAP visualization of (*d* = 2000) genes where each point indicates a dynamic or static gene identified by the model. **c** A permutation test was performed using experimentally-derived association scores from the STRING database[122] to assess whether genes within DELVE dynamic modules had evidence of co-regulation. (**c** top) STRING networks, where nodes represent genes from a DELVE module and edges represent experimental evidence of association. (c middle) Average pairwise change in expression amongst genes within a module ordered by induction time. Error bands represent the standard deviation. (**c** bottom) Histograms showing the distribution of the average degree of experimentally-derived edges of gene networks from *R* = 1000 random permutations. The dotted line indicates the observed average degree from

genes within a DELVE module. *p* values were computed using a one-sided permutation test. **d** Heatmap visualization of the expression of dynamically-expressed genes identified by DELVE ordered across estimated pseudotime. A subset of genes were labeled for readability. **e** PHATE visualizations of the DE differentiation trajectory for three feature selection approaches. Barplots show the mean ± standard deviation for the top 15 gene set enrichment terms associated with the temporally-expressed gene lists specific to each feature selection strategy following trajectory inference with *n* = 10 random root cells. *p*-values were computed using a Fisher exact test and adjusted with Benjamini Hochberg correction. **f** The venn diagram illustrates the overlap of DE genes across feature selection strategies. **g** Barplots indicate mean ± standard deviation of the number of transcription factors identified following trajectory inference using *n* = 10 random root cells for each feature selection approach. **h** Pseudotime traces of example hESC, PS, or DE-specific transcription factors identified by DELVE and missed by RNA velocity. Source data are provided in a Source Data file.

or variance-based feature selection methods on identifying biologically-relevant features.

Next, we applied DELVE to a variety of biological contexts and demonstrated improved recovery of cellular trajectories over existing unsupervised feature selection strategies. Specifically, in the context of studying the cell cycle from protein imaging data[17], we illustrated how DELVE identified proteomic imaging-derived features that were strongly associated with cell cycle progression and were more

biologically predictive of cell cycle phase and age, as compared to the alternative unsupervised feature selection methods. Importantly, DELVE often achieved similar or better performance to the supervised Random Forest classification approach without the need for training on ground truth cell cycle labels. Moreover, we illustrated how leveraging a bottom-up approach with DELVE was crucial for parsing cell cycle progression-specific features from noisy extracted imaging measurements profiled with 4i, where DELVE often outperformed the

alternative unsupervised similarity-based feature selection methods Laplacian score and Hotspot. In the context of studying heterogeneous CD8+ T cell response to viral infection from single-cell RNA sequencing data[107], we showed how DELVE identified gene complexes that had experimental evidence of co-regulation and were strongly associated with CD8+ T cell differentiation. Furthermore, we showed how performing feature selection with DELVE prior to performing trajectory inference improved the identification and resolution of gene programs associated with long-term memory formation that would have been missed by the standard unsupervised highly variable gene selection. Lastly, in the context of studying human embryonic stem cell differentiation into the definitive endoderm from single-cell RNA sequencing data, we showed how DELVE improved the recovery of gene programs involved in pluripotency, gastrulation, and definitive endoderm specification, and identified more known transcription factors and master regulators missed by HVG and RNA velocity.

This study highlights how DELVE can be used to improve inference of cellular trajectories in the context of noisy single-cell omics data; however, it is important to note that feature selection can greatly bias the interpretation of the underlying cellular trajectory[45], thus careful consideration should be made when performing feature selection for trajectory analysis. Furthermore, we provided an unsupervised framework for ranking features according to their association with temporally co-expressed features, although we note that DELVE can be improved by using a set of previously established regulators. Moreover, in this study we leveraged all dynamic modules equally for constructing the approximate cellular trajectory; however, we note that is possible to extend this framework to exclude unwanted dynamic modules of interest prior to ranking feature performance. Future work could focus on extending this framework for applications such as (1) deconvolving cellular trajectories using biological system-specific seed graphs or (2) studying complex biological systems such as organoid models or spatial microenvironments.

## Methods
### DELVE
DELVE identifies a subset of dynamically-changing features that preserve the local structure of the underlying cellular trajectory. In this section, we will (1) describe computational methods for the identification and ranking of features that have non-random patterns of dynamic variation, (2) explain DELVE's relation to previous work, and (3) provide context for the mathematical foundations behind discarding information-poor features prior to performing trajectory inference.

**Problem formulation.** Let $\mathbf{X} = \{\mathbf{x}_i\}_{i=1}^{n}$ denote a single-cell dataset, where $\mathbf{x}_i \in \mathbb{R}^d$ represents the vector of $d$ measured features (e.g., genes or proteins) measured in cell $i$. We assume that the data have an inherent trajectory structure, or biologically-meaningful ordering, that can be directly inferred by a limited subset of $p$ features where $p \ll d$. Therefore, our goal is to identify this limited set of $p$ features from the original high-dimensional feature set that best approximate the transitions of cells through each stage of the underlying dynamic process.

### Step 1: Dynamic seed selection
**Graph construction.** Our approach DELVE extends previous similarity-based[29,55,56] or subspace-learning[58] feature selection methods by computing the dependence of each gene on the underlying cellular trajectory. In step 1, DELVE models cell states using a weighted $k$-nearest neighbor affinity graph of cells ($k = 10$), where nodes represent cells and edges describe the transcriptomic or proteomic similarity amongst cells according to the $d$ profiled features encoded in $\mathbf{X}$. More specifically, let $\mathcal{G} = (\mathcal{V}, \mathcal{E})$ denote a between-cell affinity graph, where $\mathcal{V}$ represents the cells and the edges, $\mathcal{E}$, are weighted according to a Gaussian kernel as,

$$w_{ij} = \begin{cases} \exp\left(-\frac{\|\mathbf{x}_{v_i} - \mathbf{x}_{v_j}\|^2}{2\sigma_i^2}\right), & \text{if } v_j \in \mathcal{N}_i \\ 0, & \text{otherwise}. \end{cases} \quad (1)$$

Here, $\mathbf{W}$ is a $n \times n$ between-cell similarity matrix, where cells $v_i$ and $v_j$ are connected with an edge with edge weight $w_{ij}$ if the cell $v_j$ is within the set of $v_i$'s neighbors, as denoted by notation $\mathcal{N}_i$. Moreover, $\sigma_i$, specific for a particular cell $i$, represents the Gaussian kernel bandwidth parameter that controls the decay of cell similarity edge weights. We chose a bandwidth parameter as the distance to the 3rd nearest neighbor as this has been shown previously in refs. 56,154 to provide reasonable decay in similarity weights.

**Identification of feature modules.** To identify groups of features with similar co-expression variation, DELVE clusters features according to changes in expression across prototypical cell neighborhoods. First, cellular neighborhoods are defined according to the average expression of each set of $k$ nearest neighbors ($\mathcal{N}_i$) as, $\mathbf{Z} = \{\mathbf{z}_i \in \mathbb{R}^d\}_{i=1}^{n}$, where each $\mathbf{z}_i = \frac{1}{k}\sum_{\mathcal{N}_i} \mathbf{x}_i$ represents the center of the $k$ nearest neighbors for cell $i$ across all measured features. Next, DELVE leverages Kernel Herding sketching[69] to effectively sample $m$ representative cell neighborhoods, or rows, from the per-cell neighbor averaged feature matrix, $\mathbf{Z}$, as $\tilde{\mathbf{Z}} = \{\tilde{\mathbf{z}}_i \in \mathbb{R}^d\}_{i=1}^{m}$. This sampling approach ensures that cellular neighborhoods are more reflective of the original distribution of cell states, while removing redundant cell states to aid in the scalability of estimating expression dynamics. DELVE then computes the average pairwise change in expression of features across representative cellular neighborhoods, $\boldsymbol{\Delta}$, as,

$$\boldsymbol{\Delta} = \frac{1}{m-1}\sum_{i=1}^{m}\left(\tilde{\mathbf{Z}} - \mathbf{j}_m \tilde{\mathbf{z}}_i^{\mathsf{T}}\right), \quad (2)$$

where $\mathbf{j}_m$ is a column vector of ones with length $m$, such that $\mathbf{j}_m \in \mathbb{R}^m$. Here, $\boldsymbol{\Delta}$ is a $m \times d$ neighborhood by feature matrix, where each row corresponds to a neighborhood and contains the average pairwise change in expression for that neighborhood across all features. Lastly, features are clustered according to the transpose of their average pairwise change in expression across the representative cellular neighborhoods, $\boldsymbol{\Delta}^{\mathsf{T}}$, using the KMeans + + algorithm[105]. In this context, each DELVE module contains a set of features with similar local changes in co-variation across cell states along the cellular trajectory.

**Dynamic expression variation permutation testing.** To assess whether modules of features have coordinated or noisy expression variation, we compare the average sample variance of features within a DELVE module to random assignment using a permutation test as follows. Let $\bar{S}_c^2(P_c)$ denote the average sample variance of the average pairwise change in expression across $m$ cell neighborhoods for the set of $p$ features (a set of features denoted as $P_c$) within a DELVE cluster $c$ as,

$$\bar{S}_c^2(P_c) = \frac{1}{|P_c|}\sum_{p=1}^{|P_c|}\sum_{i=1}^{m}\frac{\left(\Delta_{i,p} - \bar{\Delta}_p\right)^2}{m-1}. \quad (3)$$

Moreover, let $R_q$ denote a set of randomly selected features sampled without replacement from the full feature space $d$, such that $|P_c| = |R_q|$, and $\tilde{S}_c^2(R_q)$ denote the average sample variance of randomly

selected feature sets averaged across $t$ random permutations as,

$$\tilde{S}_c^2(R_q) = \frac{1}{t}\sum_{q=1}^{t}\bar{S}_c^2(R_q). \tag{4}$$

Here, DELVE considers a module of features as being dynamically-expressed if the average sample variance of the change in expression of the set of features within a DELVE cluster (or specifically feature set $P_c$), is greater than random assignment, $R_q$, across randomly permuted trials as,

$$\bar{S}_c(P_c) > \tilde{S}_c(R_q). \tag{5}$$

In doing so, this approach is able to identify and exclude modules of features that have static, random, or noisy patterns of expression variation, while retaining dynamically expressed features for ranking feature importance. Given that noisy or irrelevant features can confound inference of the underlying cellular trajectory[45,46] and have been shown to corrupt the graph Laplacian for feature selection[61,62], this exclusion step is crucial prior to performing for feature selection. Of note, given that KMeans++ clustering is used to initially assign features to a group, feature-to-cluster assignments can tend to vary due to algorithm stochasticity. Therefore, to reduce the variability and find a core set of features that are consistently dynamically-expressed, this process is repeated across ten random clustering initializations and the set of dynamically-expressed features are defined as the intersection across runs.

**Step 2: Feature ranking.** Following dynamic seed selection, in step two, DELVE ranks features according to their association with the underlying cellular trajectory graph. First, DELVE approximates the underlying cellular trajectory by constructing a between-cell affinity graph, where the nodes represent the cells and edges are now re-weighted according to a Gaussian kernel between all cells based on the core subset of dynamically expressed regulators from step 1, such that $\tilde{\mathbf{X}} = \{\tilde{\mathbf{x}}_i \in \mathbb{R}^p\}$ where $p \ll d$ as,

$$\tilde{w}_{ij} = \begin{cases} \exp\left(-\frac{\|\tilde{\mathbf{x}}_{v_i} - \tilde{\mathbf{x}}_{v_j}\|^2}{2\sigma_i^2}\right), & \text{if } v_j \in \mathcal{N}_i \\ 0, & \text{otherwise} \end{cases} \tag{6}$$

Here, $\tilde{\mathbf{W}}$ is a $n \times n$ between-cell similarity matrix, where cells $v_i$ and $v_j$ are connected with an edge with edge weight $\tilde{w}_{ij}$ if the cell is within the set of $v_i$'s neighbors, denoted as $\mathcal{N}_i$. Moreover, as previously mentioned, $\sigma_i$ represents the Gaussian kernel bandwidth parameter for a particular cell $i$ as the distance to the 3rd nearest neighbor.

Features are then ranked according to their association with the underlying cellular trajectory graph using graph signal processing techniques[55,70,71]. By modeling data as a set of signals on a weighted graph, graph signal processing techniques have led to the development of new machine learning models for improved biological prediction, including regression or classification[70,155], identification of prototypical cells associated with experimental perturbations[82], or inference of cell signaling dynamics[156]. A graph signal $f$ is any function that has a real defined value on all of the nodes, such that $f \in \mathbb{R}^n$ and $f_i$ gives the signal at the $i$th node. Intuitively, in DELVE, we consider all features as graph signals and rank them according to their variation in expression along the approximate cell trajectory graph to see if they should be included or excluded from downstream analysis. Let $\mathbf{L}$ denote the unnormalized graph Laplacian, with $\mathbf{L} = \mathbf{D} - \tilde{\mathbf{W}}$, where $\mathbf{D}$ is a diagonal degree matrix with each element as $d_{ii} = \sum_j \tilde{w}_{ij}$. The local variation in the expression of feature signal $\mathbf{f}$ can then be defined as the weighted sum of differences in signals around a particular cell $i$ as,

$$(\mathbf{L}\mathbf{f})(i) = \sum_j \tilde{\mathbf{w}}_{ij}(\mathbf{f}(i) - \mathbf{f}(j)). \tag{7}$$

This metric effectively measures the similarity in expression of a particular node's graph signal, denoted by the feature vector, $\mathbf{f}$, around its $k$ nearest neighbors. By summing the local variation in expression across all neighbors along the cellular trajectory, we can define the total variation in expression of feature graph signal $\mathbf{f}$ as,

$$\mathbf{f}^\mathsf{T}\mathbf{L}\mathbf{f} = \sum_{ij} \tilde{\mathbf{w}}_{ij}(\mathbf{f}(i) - \mathbf{f}(j))^2. \tag{8}$$

Otherwise known as the Laplacian quadratic form[71,157,158], in this context, the total variation represents the global smoothness of the particular graph signal encoded in $\mathbf{f}$ (e.g., expression of a particular gene or protein) along the approximate cellular trajectory graph. Intuitively, DELVE aims to retain features that have a low total variation in expression, or have similar expression values amongst similar cells along the approximate cellular trajectory graph. In contrast, DELVE excludes features that have a high total variation in expression, or those which have expression values that are rapidly oscillating amongst neighboring cells, as these features are likely noisy or not involved in the underlying dynamic process that was initially seeded.

In this work, we ranked features according to their association with the cell-to-cell affinity graph defined by a core set of dynamically expressed regulators from DELVE dynamic modules using the Laplacian score[55]. This measure takes into account both the total variation in expression, as well as the overall global variance. For each of the original $d$ measured features, or *graph signals* encoded in $\mathbf{f}$ with $\mathbf{f} \in \mathbb{R}^n$, the Laplacian score $L_f$ is computed as,

$$L_f = \frac{\tilde{\mathbf{f}}^\mathsf{T}\mathbf{L}\tilde{\mathbf{f}}}{\tilde{\mathbf{f}}^\mathsf{T}\mathbf{D}\tilde{\mathbf{f}}}. \tag{9}$$

Here, $\mathbf{L}$ represents the unnormalized graph Laplacian, such that $\mathbf{L} = \mathbf{D} - \tilde{\mathbf{W}}$, $\mathbf{D}$ is a diagonal degree matrix with the $i$th element of the diagonal $d_{ii}$ as $d_{ii} = \sum_j \tilde{w}_{ij}$, $\tilde{\mathbf{f}}$ represents the mean centered expression of feature $\mathbf{f}$ as $\tilde{\mathbf{f}} = \mathbf{f} - \frac{\mathbf{f}^\mathsf{T}\mathbf{D}\mathbf{1}}{\mathbf{1}^\mathsf{T}\mathbf{D}\mathbf{1}}$, and $\mathbf{1} = [1, ..., 1]^\mathsf{T}$. By sorting features in ascending order according to their Laplacian score, DELVE effectively ranks features that best preserve the local trajectory structure (e.g., an ideal numerator has a small local variation in expression along neighboring cells), as well as best preserve cell types (e.g., an ideal denominator has large variance in expression for discriminitive power).

**Guidelines on parameter selection.** In this section, we provide practical recommendations for selecting hyperparameters in DELVE and describe DELVE's sensitivity to choices in hyperparameters for both simulated single-cell RNA sequencing data and proteomic imaging data. We recommend standard quality control preprocessing prior to performing feature selection.

**Number of nearest neighbors ($k$).** DELVE uses a $k$-nearest neighbor between-cell affinity graph to (1) identify dynamic modules of features and (2) rank features according to their association with the underlying cellular trajectory. The choice of number of nearest neighbors, $k$, in the $k$-nearest neighbor graph construction directly influences the connectivity of the graph. Selecting a small value of $k$ prioritizes local relationships amongst cells, whereas selecting a large value of $k$ connects more dissimilar cells together and prioritizes global relationships. Since DELVE ranks features by summing the differences in feature expression around nearest neighbors along the cellular

trajectory graph (See Equation 8), the choice of number of nearest neighbors, $k$, may influence the overall resolution of feature dynamics and feature ranking. Overall, we recommend selecting a small number of nearest neighbors, such that the local connectivity of similar cell states are preserved. Moreover, we recommend constructing a $k$-nearest neighbors graph based on Euclidean distances in principal component space for single-cell RNA sequencing datasets that have not been variance-filtered or contain many more genes than cells, $p \gg n$[72,159].

**Number of subsampled neighborhoods ($m$).** We have previously shown that selecting a small representative subset of cells with Kernel Herding sketching (1) preserves the original distribution of cell states, (2) achieves the same fidelity of downstream performance with far fewer cells, and (3) results in improved scalability and faster runtimes[69]. However, it is important to note that selecting too small of a sketch size (i.e., too few subsampled neighborhoods, $m \ll n$) has the potential to neglect rare cell states. This may impact the identification of associated feature dynamics. Therefore, we recommend selecting a larger number of subsampled neighborhoods from the original dataset to retain expected rare cell populations that occur at a small frequency.

**Number of modules ($c$).** To identify modules of features similar expression dynamics, DELVE clusters features according to their average pairwise change in expression across prototypical cellular neighborhoods using the $K$-means++ algorithm. The number of clusters, $c$, should be chosen according to the number and granularity of desired dynamic and static modules. Selecting too few clusters (e.g., $c < 3$) can result in modules of features with more heterogeneity in their expression and/or retention of correlated noisy features. Moreover, as the number of clusters becomes very large proportionally to the total number of features, clusters will contain far fewer features, which may result in reduced power to detect grouped feature dynamics in the permutation test. Of note, for particularly noisy imaging datasets with many correlated features (e.g., PDAC 4i cell cycle), we found that selecting more modules ($c = 10$) improved the separation and retention of a core set of dynamic cell cycle features from noisy correlated modules.

**DELVE's sensitivity to parameter choices.** To evaluate DELVE's sensitivity to choices in hyperparameters, we varied the number of neighbors per cell ($k = 5, 10, 20, 30, 40, 50, 60, 70, 80, 90, 100$), the number of subsampled neighborhoods ($m = 250, 500, 1000, 1500, 2000$), and the number of KMeans++ clusters in dynamic module identification ($c = 3, 4, 5, 6, 7, 8, 9, 10$). Feature selection method performance was assessed on datasets with ground truth reference information (i.e., 4i RPE cell cycle dataset using 30 selected proteomic features, SymSim simulated single-cell RNA sequencing dataset using 1000 selected genes) by computing several metrics of trajectory preservation, including (1) $k$ nearest neighbor classification accuracy between predicted and ground truth cell type labels, (2) NMI clustering score between predicted and ground truth cell type labels, and (3) Kendall rank correlation between estimated pseudotime and the ground truth measurements using either the diffusion pseudotime algorithm[31] or Slingshot[28] for trajectory inference. Results were computed across 20 random trials for each dataset and parameter configuration. Overall, we found DELVE was robust to choices in hyperparameters (Supplementary Figs. 14, 15). However, we note that parameter choices should be selected according to the profiled dataset.

**Benchmarked feature selection methods**
In this section, we describe the twelve feature selection methods evaluated for representing biological trajectories. For more details on

implementation and hyperparameters, see Supplementary Table 1 and Guidelines on parameter selection.

**Random forest.** To quantitatively compare feature selection approaches on preserving biologically relevant genes or proteins, we aimed to implement an approach that would leverage ground truth cell type labels to determine feature importance. Random forest classification[49] is a supervised ensemble learning algorithm that uses an ensemble of decision trees to partition the feature space such that all of the samples (cells) with the same class (cell type labels) are grouped together. Each decision or split of a tree was chosen by minimizing the Gini impurity score as,

$$G(m) = \sum_{i=1}^{c} p_{mi}(1 - p_{mi}). \tag{10}$$

Here, $p_{mi}$ is the proportion of cells that belong to class $i$ for a feature node $m$, and $C$ is the total number of classes (e.g., cell types). We performed random forest classification using the scikit-learn v0.23.2 package in python. Nested 10-fold cross-validation was performed using stratified random sampling to assign cells to either a training or test set. The number of trees was tuned over a grid search within each fold prior to training the model. Feature importance scores were subsequently determined by the average Gini importance across folds.

**Max variance.** Max variance is an unsupervised feature selection approach that uses sample variance as a criterion for retaining discriminative features, where $\hat{S}_f^2$ represents the sample variance for feature $\mathbf{f} \in \mathbb{R}^n$ as,

$$S_f^2 = \frac{1}{n-1} \sum_{i=1}^{n} (f_i - \bar{f})^2, \tag{11}$$

where $f_i$ indicates the expression value of feature $\mathbf{f}$ in cell $i$. We performed max variance feature selection by sorting features in descending order according to their variance score and selecting the top $p$ maximally varying features.

**Neighborhood variance.** Neighborhood variance[29] is an unsupervised feature selection approach that uses a local neighborhood variance metric to select gradually-changing features for building biological trajectories. Namely, the neighborhood variance metric $\tilde{S}_f^2$ quantifies how much feature $f$ varies across neighboring cells as,

$$\tilde{S}_f^2 = \frac{1}{nk_c - 1} \sum_{i=1}^{n} \sum_{j=1}^{k_c} (f_i - f_{\mathcal{N}_{(i,j)}})^2. \tag{12}$$

Here, $f_i$ represents the expression value of feature $f$ for cell $i$, $\mathcal{N}_{(i,j)}$ indicates the $j$ nearest neighbor of cell $i$, and $k_c$ is the minimum number of $k$-nearest neighbors required to form a fully connected graph. Features were subsequently selected if they had a smaller neighborhood variance $\tilde{S}_f^2$ than global variance $S_f^2$,

$$\frac{S_f^2}{\tilde{S}_f^2} > 1. \tag{13}$$

**Highly variable genes.** Highly variable gene selection[53] is an unsupervised feature selection approach that selects features according to a normalized dispersion measure. First, features are binned based on their average expression. Within each bin, genes are then z-score normalized to identify features that have a large variance, yet a similar mean expression. We selected the top $p$ features using the highly

variable genes function in Scanpy v1.9.3 (flavor = Seurat, bins = 20, n_top_genes = $p$).

**Laplacian score.** Laplacian score (LS)[55] is a locality-preserving unsupervised feature selection method that ranks features according to (1) how well a feature's expression is consistent across neighboring cells defined by a between-cell similarity graph define by all profiled features and (2) the feature's global variance. First, a weighted $k$-nearest neighbor affinity graph of cells ($k = 10$) is constructed according to pairwise Euclidean distances between cells based on all features, **X**. More specifically, let $\mathcal{G} = (\mathcal{V}, \mathcal{E})$, where $\mathcal{V}$ represents the cells and edges, $\mathcal{E}$, are weighted using a Gaussian as follows. Specifically, edge weights between cells $i$ and $j$ can be defined as,

$$w_{ij} = \begin{cases} \exp\left(-\frac{\|\mathbf{x}_{v_i} - \mathbf{x}_{v_j}\|^2}{2\sigma_i^2}\right), & \text{if } v_j \in \mathcal{N}_i \\ 0, & \text{otherwise}. \end{cases} \quad (14)$$

Here **W** is a $n \times n$ between-cell similarity matrix, where cells $v_i$ and $v_j$ are connected with an edge with edge weight $w_{ij}$ if the cell $v_j$ is within the set of $v_i$'s neighbors, $\mathcal{N}_i$. Moreover, as previously described, $\sigma_i$ represents the bandwidth parameter for cell $i$ defined as the distance to the 3rd nearest neighbor. For each feature **f**, where $\mathbf{f} \in \mathbb{R}^n$ represents the value of the feature across all $n$ cells, we compute the Laplacian score, $L_f$ as,

$$L_f = \frac{\tilde{\mathbf{f}}^T \mathbf{L} \tilde{\mathbf{f}}}{\tilde{\mathbf{f}}^T \mathbf{D} \tilde{\mathbf{f}}}. \quad (15)$$

Here, **L** represents the unnormalized graph Laplacian, with $\mathbf{L} = \mathbf{D} - \mathbf{W}$, **D** is a diagonal degree matrix with the $i$th element of the diagonal $d_{ii}$ as $d_{ii} = \sum_j w_{ij}$, $\tilde{\mathbf{f}}$ represents the mean centered expression of feature **f** as $\tilde{\mathbf{f}} = \mathbf{f} - \frac{\mathbf{f}^T \mathbf{D} \mathbf{1}}{\mathbf{1}^T \mathbf{D} \mathbf{1}}$, and $\mathbf{1} = [1, \dots, 1]^T$. We performed feature selection by sorting features in ascending order according to their Laplacian score and selecting the top $p$ features.

**MCFS.** Multi-cluster feature selection (MCFS)[58] is an unsupervised feature selection method that selects for features that best preserve the multi-cluster structure of data by solving an L1 regularized least squares regression problem on the spectral embedding. Similar to the Laplacian score, first $k$-nearest neighbor affinity graph of cells ($k = 10$) is computed to encode the similarities in feature expression between cells $i$ and $j$ using a Gaussian kernel as,

$$w_{ij} = \begin{cases} \exp\left(-\frac{\|\mathbf{x}_{v_i} - \mathbf{x}_{v_j}\|^2}{2\sigma_i^2}\right), & \text{if } v_j \in \mathcal{N}_i \\ 0, & \text{otherwise}. \end{cases} \quad (16)$$

Similar to previous formulations above, **W** is an $n \times n$ between cell affinity matrix, where a pair of cells $v_i$ and $v_j$ are connected with an edge with weight $w_{ij}$ if cell $v_j$ is within the set of $v_i$'s neighbors, $\mathcal{N}_i$. Further, $\sigma_i$ represents the kernel bandwidth parameter chosen to be the distance to the third nearest neighbor from cell $i$. Next, to represent the intrinsic dimensionality of the data, the spectral embedding[160] is computed through eigendecomposition of the unnormalized graph Laplacian **L**, where $\mathbf{L} = \mathbf{D} - \mathbf{W}$ as,

$$\mathbf{L}\mathbf{y} = \lambda \mathbf{D}\mathbf{y}. \quad (17)$$

Here, $\mathbf{Y} = \{\mathbf{y}\}_{k=1}^K$ are the eigenvectors corresponding to the $K$ smallest eigenvalues, **W** is a symmetric affinity matrix encoding cell similarity weights, and **D** represents a diagonal degree matrix with each element as $d_{ii} = \sum_j w_{ij}$. Given that eigenvectors of the graph Laplacian represent frequency harmonics[71] and low frequency eigenvectors are considered to capture the informative structure of the

data, MCFS computes the importance of each feature along each intrinsic dimension $\mathbf{y}_k$ by finding a relevant subset of features by minimizing the error using an L1 norm penalty as,

$$\min_{\mathbf{a}_k} \|\mathbf{y}_k - \mathbf{X}^T \mathbf{a}_k\|^2 \quad \text{s.t.} \quad \|\mathbf{a}_k\|_1 \leq \gamma. \quad (18)$$

Here, the non-zero coefficients, $\mathbf{a}_k$, indicate the most relevant features for distinguishing clusters from the embedding space, $\mathbf{y}_k$ and $\gamma$ controls the sparsity and ensures the least relevant coefficients are shrunk to zero. The optimization is solved using the least angles regression algorithm[74], where for every feature, the MCFS score is defined as,

$$\text{MCFS}(j) = \max_k \|a_{k,j}\|. \quad (19)$$

Here, $j$ and $k$ index feature and eigenvector, respectively. We performed multi-cluster feature selection with the number of eigenvectors $K$ chosen to be the number of ground truth cell types present within the data, as this is the traditional convention in spectral clustering[60] and the number of nonzero coefficients was set to the number of selected features, $p$.

**SCMER.** Single-cell manifold-preserving feature selection (SCMER)[57] selects a subset of $p$ features that represent the embedding structure of the data by learning a sparse weight vector **w** by formulating an elastic net regression problem that minimizes the KL divergence between a cell similarity matrix defined by all features and one defined by a reduced subset of features. More specifically, let **P** denote a between-cell pairwise similarity matrix defined in UMAP[59] computed with the full data matrix $\mathbf{X} \in \mathbb{R}^{n \times d}$ and **Q** denote a between-cell pairwise similarity matrix defined in UMAP computed with the dataset following feature selection $\mathbf{Y} \in \mathbb{R}^{n \times p}$, where $\mathbf{Y} = \mathbf{X}\mathbf{w}$ and $p \ll d$. Here, elastic net regression is used to find a sparse and robust solution of **w** that minimizes the KL divergence as,

$$\text{KL}(\mathbf{P} \| \mathbf{Q}) = \sum_i \sum_j p_{ij} \log \frac{p_{ij}}{q_{ij}}. \quad (20)$$

Features with non-zero weights in **w** are considered useful for preserving the embedding structure and selected for downstream analysis. We performed SCMER feature selection using the scmer v.0.1.0a3 package in python by constructing a $k$-nearest neighbor graph ($k = 10$) according to pairwise Euclidean distances of cells based on their first 50 principal components and using the default regression penalty weight parameters (lasso = $3.87e - 4$, ridge = 0).

**Hotspot.** Hotspot[56] is an unsupervised gene module identification approach that performs feature selection through a test statistic that measures the association of a gene's expression with the between-cell similarity graph defined based on the full feature matrix, **X**. More specifically, first, a $k$-nearest neighbor cell affinity graph ($k = 10$) is defined based on pairwise Euclidean distances between all pairs of cells using a Gaussian kernel as,

$$w_{ij} = \begin{cases} \exp\left(-\frac{\|\mathbf{x}_{v_i} - \mathbf{x}_{v_j}\|^2}{\sigma_i^2}\right), & \text{if } v_j \in \mathcal{N}_i \\ 0, & \text{otherwise}. \end{cases} \quad (21)$$

Here, cells $v_i$ and $v_j$ are connected with an edge with edge weight $w_{ij}$ if the cell $v_j$ is within the set of $v_i$'s neighbors such that $\sum_j w_{ij} = 1$ for each cell and $\sigma_i$ represents the bandwidth parameter for cell $i$ defined as the distance to the $\frac{k}{3}$ neighbor. For a given feature $\mathbf{f} \in \mathbb{R}^n$, representing expression across all $n$ cells where $f_i$ is the mean-centered and standardized expression of feature **f** in cell $i$ according to a null distribution model of gene expression, the local autocorrelation test

statistic representing the dependence of each gene on the graph structure is defined as,

$$H_f = \sum_i \sum_{j \neq i} w_{ij} f_i f_j. \tag{22}$$

Hotspot was implemented using the hotspotsc v1.1.1 package in python, where we selected the top $p$ features by sorting features in ascending order according to their significance with respect to a null model. For the single-cell RNA sequencing datasets where count data were available (splatter simulation, SymSim simulation, DE differentiation datasets), or for the the indirect immunofluorescence imaging datasets (RPE, PDAC), the null model of counts was defined by a negative binomial distribution. For the single-cell RNA sequencing datasets where only log normalized data were available (CD8 differentiation), the null model of normalized counts was defined as a normal distribution.

**All features.** To consider a baseline representation without feature selection, we evaluated performance using all features from each dataset following quality control preprocessing.

**Random features.** As a second baseline strategy, we simply selected a subset of random features without replacement. Results were computed across twenty random initializations for each dataset.

**DELVE.** DELVE was run as previously described. Here, we constructed a weighted $k$-nearest neighbor affinity graph of cells ($k = 10$), and 1000 neighborhoods were sketched to identify dynamic seed feature clusters ($c = 3$ for the Splatter simulated datasets, $c = 5$ for the SymSim simulated, RPE cell cycle, CD8 T cell differentiation, and DE differentiation datasets, or $c = 10$ for the PDAC cell cycle datasets). For practical guidelines on parameter selection, see Guidelines on parameter selection.

### Datasets

We evaluated feature selection methods based on how well retained features could adequately recover biological trajectories under various noise conditions, biological contexts, and single-cell technologies.

**Splatter simulation.** Splatter[75] is a single-cell RNA sequencing simulation software that generates count data using a gamma-Poisson hierarchical model with modifications to alter the mean-variance relationship amongst genes, the library size, or sparsity. We used splatter to simulate a total of 90 ground truth datasets with different trajectory structures (e.g., 30 linear datasets, 30 bifurcation datasets, and 30 tree datasets). First, we estimated simulation parameters by fitting the model to a real single-cell RNA sequencing dataset consisting of human pluripotent stem cells differentiating into mesoderm progenitors[138]. We then used the estimated parameters (mean_rate = 0.0173, mean_shape = 0.54, lib_loc = 12.6, lib_scale = 0.423, out_prob = 0.000342, out_fac_loc = 0.1, out_fac_scale = 0.4, bcv = 0.1, bcv_df = 90.2, dropout = None) to simulate a diverse set of ground truth reference trajectory datasets with the splatter paths function (python wrapper scprep SplatSimulate v1.2.3 of splatter v1.18.2). Here, a reference trajectory structure (e.g., bifurcation) was used to simulate linear and nonlinear changes in the mean expression of genes along each step of the specified differentiation path. We simulated differentiation datasets (1500 cells, 500 genes, 6 clusters) for each trajectory type (linear, bifurcation, tree) by modifying (1) the probability of a cell belonging to a cluster by randomly sampling from a Dirichlet distribution with six categories and a uniform concentration of one and (2) the path skew by randomly sampling from a beta distribution ($\alpha = 10, \beta = 10$). The output of each simulation is a ground truth reference consisting of cell-to-cluster membership, differentially expressed genes per

cluster or path, as well as a latent *step* vector that indicates the progression of each cell within a cluster. Lastly, we modified the step vector to be monotonically increasing across clusters within the specified differentiation path to obtain a reference pseudotime measurement.

To estimate how well feature selection methods can identify genes that represent cell populations and are differentially expressed along a differentiation path in noisy single-cell RNA sequencing data, we added relevant sources of biological and technical noise to the reference datasets.

1. Biological Coefficient of Variation (BCV): To simulate the effect of stochastic gene expression, we modified the biological coefficient of variation parameter within splatter (BCV = 0.1, 0.25, 0.5). This scaling factor controls the mean-variance relationship between genes, where lowly expressed genes are more variable than highly expressed genes, following a $\gamma$ distribution. This corresponded to an approximate mean coefficient of variation 1.55, 1.60, and 1.75 when averaged across all genes and max coefficient of variation of 2.90, 2.95, and 3.10.

2. Library size: The total number of profiled mRNA transcripts per cell, or library size, can vary between cells within a single-cell RNA sequencing experiment and can influence the detection of differentially expressed genes[76], as well as impact the reproducibility of the lower-dimensional representation of the data[77]. To simulate the effect of differences in sequencing depth, we proportionally adjusted the gene means for each cell by modifying the location parameter (lib_loc = 12, 11, 10) of the log-normal distribution within splatter that estimates the library size scaling factors. This corresponded to an average library size of approximately $3e^4, 1.7e^4$ and $9e^3$.

3. Technical dropout: Single-cell RNA sequencing data contain a large proportion of zeros, where only a small fraction of total transcripts are detected due to capture inefficiency and amplification noise[161]. To simulate the inefficient capture of mRNA molecules and account for the trend that lowly expressed genes are more likely to be affected by dropout, we undersampled mRNA counts by sampling from a binomial distribution with the scale parameter or dropout rate proportional to the mean expression of each gene as previously described in ref. 162 as,

$$r_i = \exp(-\lambda \mu_i^2). \tag{23}$$

Here, $\mu_i$ represents the log mean expression of gene $i$, and $\lambda$ is a hyperparameter that controls the magnitude of dropout ($\lambda = 0, 0.05, 0.1$). This corresponded to an approximate undersampling percentage of 0, 10, and 20 percent when averaged across all cells for all genes.

In our subsequent feature selection method analyses, we selected the top $p = 100$ features under each feature selection approach.

**SymSim simulation.** SymSim[86] is a single cell RNA sequencing simulation software that uses a kinetic model of gene expression followed by library preparation and sequencing simulation to model intrinsic, extrinsic, and technical variability in single-cell RNA sequencing data. This approach was shown in a recent benchmarking study[87] to be amongst the top ranking methods for reasonably simulating single-cell RNA sequencing data as measured by their accuracy in estimating data properties (e.g., library size, TMM, mean expression, scaled variance, fractions of zeros, cell and gene correlations) and their ability to preserve biological signals (e.g., differentially expressed genes, differentially variable genes). We used SymSim v0.0.0.9000 in R v4.1.1 to simulate tree differentiation trajectories with 10000 cells, 20000 genes, and 4 cell types. Simulation parameters ($\sigma = 0.6, 0.4, \alpha_{\text{mean}} = (0.01, 0.02, 0.03, 0.04, 0.05), \alpha_{\text{sd}} = 0.02$, evf_type = continuous,

`protocol = UMI, vary = s, nevf = 1.5% total genes, n_de_evf = 1% · total genes, depth_mean = 5e^5, depth_sd = 3e^4`) were chosen from Table 2 in ref. 87, as this table contained simulation parameters that achieved the most similar distributions of statistics to real single-cell RNA sequencing data. To evaluate how well feature selection methods could identify genes that define cellular trajectories when subjected to a reduction of the total mRNA count, we performed two experiments where we simulated five differentiation trajectories by modifying the mean mRNA capture efficiency rate $\alpha_{mean} = (0.01, 0.02, 0.03, 0.04, 0.05)$ for cells with a high within population variability ($\sigma = 0.6$) and low within population variability ($\sigma = 0.4$). The output of each simulated dataset was a ground truth reference trajectory containing cell-to-cluster membership, as well as a latent pseudotime vector that indicated cell progression along each population in the trajectory. In our subsequent feature selection method analyses, we first performed principal components analysis (`n_pcs = 50`) and then selected the top $p = 2000, 1000, 500, 250, 100,$ or $50$ features under each feature selection approach.

**RPE analysis.** The retinal pigmented epithelial (RPE) dataset[17] is an iterative indirect immunofluorescence imaging (4i) dataset consisting of RPE cells undergoing the cell cycle. Here, time-lapse imaging was performed on an asynchronous population of non-transformed human retinal pigmented epithelial cells expressing a PCNA-mTurquoise2 reporter in order to record the cell cycle phase (G0/G1, S, G2, M) and age (time since last mitosis) of each cell. Following time-lapse imaging, cells were fixed and 48 core cell cycle effectors were profiled using 4i[8]. For preprocessing, we min-max normalized the data and performed batch effect correction on the replicates using the ComBat[163] function in Scanpy v1.9.3. Lastly, to refine phase annotations and distinguish G0 from G1 cells, we selected cycling G1 cells according to the bimodal distribution of $^{pRB}/_{RB}$ expression as described in ref. 17. Of note, cells were excluded if they did not have ground truth phase or age annotations. The resultant dataset consisted of 2759 cells × 241 imaging-derived features describing the expression and localization of different protein markers. For our subsequent analysis, we selected the top $p = 30$ imaging-derived features for each feature selection approach.

**PDAC analysis.** The pancreatic ductal adenocarcinoma (PDAC) cell cycle dataset[164] is an iterative indirect immunofluorescence imaging dataset that profiled 63 core cell cycle effectors in 9 human PDAC cell lines: BxPC3, CFPAC, HPAC, MiaPaCa, Pa01C, Pa02C, Pa16C, PANC1, UM53. Each dataset resulted in $d = 253$ imaging-derived features representing the expression and localization of different protein markers. For each cell line (e.g., BxPC3 under control conditions), we min-max normalized the data. Cell cycle phases (G0, G1, S, G2, M) were annotated a priori based on manual gating cells according to the abundance of core cell cycle markers. Phospho-RB (pRB) was used to distinguish proliferative cells (G1/S/G2/M, high pRB) from arrested cells (G0, low pRB). DNA content, E2F1, cyclin A (cycA), and phospho-p21 (p-p21) were used to distinguish G1 (DNA content = 2C, low cycA), S (DNA content = 2-4C, high E2F1), G2 (DNA content = 4C, high cycA), and M (DNA content = 4C, high p-p21). For our subsequent analysis, we selected the top $p = 30$ features for each feature selection approach.

**CD8+ T cell differentiation analysis.** The CD8+ T cell differentiation dataset[107] is a single-cell RNA sequencing dataset consisting of mouse splenic CD8+ T cells profiled over 12-time points ($d$ = day) following infection with the Armstrong strain of the lymphocytic choriomeningitis virus: Naive, d3-, d4-, d5-, d6-, d7-, d10-, d14-, d21-, d32-, d60-, d90- post-infection. Spleen single-cell RNA sequencing data were accessed from the Gene Expression Omnibus using the accession code GSE131847 and concatenated into a single matrix. The dataset was subsequently quality control filtered according to the distribution of molecular counts. To remove dead or dying cells, we filtered cells that had more than twenty

percent of their total reads mapped to mitochondrial transcripts. Genes that were observed in less than three cells or had less than 400 counts were also removed. Following cell and gene filtering, the data were transcripts-per-million normalized, log+1 transformed, and variance filtered using highly variable gene selection, such that the resulting dataset consisted of 29893 cells × 2000 genes (See Highly variable genes). When evaluating feature selection methods, we first performed principal components analysis $n\_pcs = 50$, and then selected the top $p = 500$ features for each feature selection approach.

**DE differentiation analysis.** The DE differentiation dataset is a multiplexed single-cell RNA sequencing dataset consisting of human embryonic stem cells (hESCs) differentiating into the DE. Cells were profiled over a 2 day time course following induction of TGF$\beta$ and WNT signaling pathways: day 0, day 1-, day2-post treatment.

1. Cell culture of human embryonic stem cells: H9 human embryonic stem cells (WiCell) were grown in the medium mTesR1 (STEMCELL Technologies) in tissue culture dishes coated with Matrigel (Corning; 1:100 in DMEM/F12) overnight at 4 °C. Media changes were performed daily. Cells were routinely passaged every 2–3 days using Dulbecco's PBS/0.5 mM EDTA and treated with ROCK inhibitor Y27672 (10 µM; STEMCELL Technologies) for 24 h after passaging. The cells were kept at 37 °C, and 5% CO2 in a tissue culture incubator.

2. Definitive endoderm differentiation: Definitive endoderm differentiation of H9 human embryonic stem cells (WiCell) was performed according by adapting the modified D'Amour protocol previously described in ref. 129. For each treatment group, 100K cells were seeded into a single well of a 12-well plate with mTeSR (Stemcell Technologies) and ROCK inhibitor Y-27632 (10 µM; STEMCELL Technologies). After 24 h, cells were switched to mTeSR without ROCK inhibitor Y-27632 for an additional 24 h. To induce definitive endoderm differentiation, cells were treated with Activin A ($100^{ng}/_{mL}$; Peprotech) and CHIR90992 (5 µM; Peprotech) in RPMI 1640 media with B27 supplement for 24 h (d1-post treatment) followed by Activin A ($100^{ng}/_{mL}$; Peprotech) in RPMI 1640 media with B27 supplement for the final 24 h (d2-post treatment). Cell seeding was staggered for simultaneous cell collection and labeling across time points.

3. Multiplexed single-cell RNA sequencing: We performed multiplexed single-cell RNA sequencing by adapting an approach previously outlined in ref. 128 that uses Click Chemistry to tag cells with unique oligo barcodes for condition-specific cell labeling and multiplexing prior to single-cell RNA sequencing. Condition-specific oligo barcodes were designed according to 10X Genomics' specifications for feature barcoding of cell surface proteins. Each condition-specific oligo barcode contained a capture sequence 1, tru seq read 2, a unique feature barcode selected from the list of whitelisted barcodes provided by 10X Genomics, and an amine group (Am6C) on the 5′ end to allow for the required click chemistry alterations. Condition-specific oligo barcodes (desalted, 250 nmol, IDT) were reconstituted in 250 µL distilled water and spun for 10 min to remove any insoluble debris from synthesis. Barcodes were then subjected to ethanol precipitation and resuspended in 100 muL 1X borate buffer (ThermoFisher Scientific). Concentrations in µM were calculated as $\left(^{A260}/_{\epsilon} \times \text{Dilution Factor} \times 10^6\right)$. Then 35 nmol was diluted to 50 µL with water and mixed with 7 muL 10X BBS (0.5 M borate + 1.5 M NaCl) and 7 µL DMSO. The barcodes were then reacted with 2 additions of 3.5 µL freshly prepared 100 mM MTZ-PEG4-NHS (Click Chemistry Tools) in DMSO added at 15 min intervals. The reaction was quenched with 1.45 µL 1 M glycine for 5 min, then ethanol precipitated and resuspended in 100 µL HEPES buffer. The concentration of each barcode was measured and samples were normalized to 100 µM. Prior to cell labeling,

cells were washed twice with HBSS (Gibco) to remove any extracellular proteins. Cells were then incubated in TCO-PEG4-NHS (Click Chemistry Tools) in HBSS for 15 min at room temperature on a rotating platform. This solution was prepared immediately prior to incubation with cells to minimize hydrolysis of NHS groups. After incubation, the TCO-PEG4-NHS solution was removed and the cells were incubated with MTZ-activated feature barcodes at ~10 μM in DMEM:F12 for an additional 20 min at 37 °C. The reaction was then quenched by incubation for 5 min with Tris HCl and MTZ-DBCO. Cells were washed once in PBS-EDTA then passaged using PBS-EDTA and collected with PBS-BSA. Cells were filtered and resuspended to a single cell suspension then equal numbers of cells from each treatment were combined and loaded onto a single lane of a 10X Genomics chip.

4. Single-cell RNA sequencing preprocessing and quality control: cDNA libraries were prepared for single-cell RNA sequencing using the 10X Genomics Version 3 platform and analyzed with the Cell Ranger pipeline v3.0.2. Raw sequencing Binary Base Call files were computationally demultiplexed by sample indices and converted into FASTQ files with the `cellranger mkfastq` function. This pipeline outputs two sets of FASTQ files, one corresponding to gene expression profiles and the other containing feature barcodes. Reads were then aligned to either the human reference transcriptome (Hg19) or the custom feature barcode reference file with the `cellranger count` function. Cells were then filtered by the inflection point on the barcode rank plot to eliminate background-associated cell barcodes. Loom files were generated using default parameters with the Velocyto v0.17 package in python. To computationally demultiplex cells following single-cell RNA sequencing, unique condition-specific oligo barcodes from the unified filtered feature-barcode matrix were median normalized to correct for differences in signal-to-noise and linked back to a sampling time point according to its maximum condition-specific feature barcode (e.g., day 0). Single-cell RNA sequencing quality control was performed according to the distributions of count depth, genes per cell, and the fraction of mitochondrial genes per cell. To remove dead or dying cells, we filtered cells that had more than twenty percent of their total reads mapped to mitochondrial transcripts. To remove empty droplets or cell doublets, cells were required to have a minimum number of 1500 transcript counts or a maximum number of 90000 transcript counts and were filtered out if they had less than 650 genes per cell or more than 7200 genes per cell. Following cell and gene filtering, the data were counts-per-million normalized, log+1 transformed, and variance filtered using highly variable gene selection, such that the resulting dataset consisted of 5397 cells × 2000 genes (See Highly variable genes).

5. RNA velocity estimation: RNA velocity[125,126] was calculated using the dynamical model implementation in ScVelo v0.2.5. More specifically, first and second order moments for velocity estimation were computed for each cell based on a $k$-nearest neighbor graph ($k = 10$) constructed according to pairwise Euclidean distances between cells using the first 50 principal components. The full dynamical model was then solved for all genes to obtain a high dimensional velocity vector for every cell. We then performed a likelihood ratio test for differential kinetics amongst the cell populations defined according to the time point labels to account for any differences in mRNA splicing or degradation kinetics. Groups of cells that exhibited different kinetic regimes were fit independently and velocity vectors were corrected.

When evaluating feature selection methods, we first performed principal components analysis, $n\_pcs = 50$, and then selected the top $p = 500$ features for each feature selection approach.

**Statistics and reproducibility.** For all publicly available datasets used in this study (CD8T, RPE, PDAC), the sample sizes, number of replicates, randomization, and blinding were determined by the original authors. No sample size calculations were performed for the RPE (ref. [17]), PDAC (ref. [164]), CD8T (ref. [107]), or DE datasets. The RPE study was performed in technical duplicates (ref. [17]), the PDAC study had a single well per cell line (ref. [164]), and the CD8T cell differentiation study collected CD8T cells that were pooled from approximately 1-6 mice at each timepoint. Here each timepoint of the analysis represented an independent experiment (ref. [107]). The RPE (ref. [17]) and PDAC (ref. [164]) studies had no randomization as there was no treatment induction for the control data. For the CD8T cell study, mice were randomly allocated into groups before adoptive transfer and mice were randomly selected for cell harvesting at specific time points (ref. [107]). For the DE dataset, cells were randomly placed into three replicate wells for each condition, treated with differentiation stimuli, and pooled prior to acquisition. There was no blinding to experimental allocation or outcome association in this study. Data exclusions were determined according to (1) quality control measures such as the distribution of molecular counts and expression of mitochondrial markers in single-cell RNA sequencing data, and/or (2) the availability of ground truth cellular annotations.

## Evaluation
### Classification and regression
***k*-nearest neighbor classification.** To quantitatively compare feature selection methods on retaining features that are representative of cell types, we aimed to implement an approach that would assess the quality of the graph structure. $k$ − nearest neighbors classification is a supervised learning algorithm that classifies data based on labels of the $k$-most similar cells according to their gene or protein expression, where the output of this algorithm is a set of labels for every cell. We performed $k$-nearest neighbors classification to predict cell type labels from simulated single-cell RNA sequencing datasets as follows. First, 3-fold cross-validation was performed using stratified random sampling to assign cells to either a training or a test set. Stratified random sampling was chosen to mitigate the effect of cell type class imbalance. Within each fold, feature selection was then performed on the training data to identify the top $p$ relevant features according to a feature selection strategy. Next, a $k$-nearest neighbor classifier ($k = 3$) was fit on the feature-selected training data to predict the cell type labels of the feature selected test query points. Here, labels were predicted as the mode of the cell type labels from the closest training data points according to Euclidean distance. Classification performance was subsequently assessed according to the median classification accuracy with respect to the ground truth cell type labels across folds. $k$-nearest neighbors classification was implemented using the scikit-learn v0.23.2 package in python.

**Support Vector Machine.** The Support Vector Machines (SVM)[165] is a supervised learning algorithm that constructs hyperplanes in the high-dimensional feature space to separate classes. We implemented SVM classification or regression using the scikit-learn v0.23.2 package in python. SVM classification was used to predict cell cycle phase labels for both RPE and PDAC 4i datasets, whereas SVM regression was used to predict age measurements from time lapse imaging for the RPE dataset. Here, Nested 10-fold cross-validation was performed using random sampling to assign cells to either a training set or a test set. Within each fold, feature selection was performed to identify the $p$ most relevant features according to a feature selection strategy. SVM hyperparameters were then tuned over a grid search and phase labels were subsequently predicted from the test data according to those $p$ features. Classification performance was assessed according to the median classification accuracy with respect to the ground truth cell type labels across folds. Regression performance was assessed

according to the average root mean squared error with respect to ground truth age measurements across folds.

**Precision@k.** To evaluate the biological relevance of selected features from each method, we computed precision@k ($p@k$) as the proportion of top $k$ selected features that were considered to be biologically relevant according to a ground truth reference as,

$$p@k = \frac{|\mathrm{F}_{s,k} \cap \mathrm{F}_r|}{|\mathrm{F}_{s,k}|}, \tag{24}$$

where $\mathrm{F}_{s,k}$ indicates the set of selected features at threshold $k$, where $\mathrm{F}_{s,k} \subset \mathrm{F}_s$, and $\mathrm{F}_r$ indicates the set of reference features. Reference features were defined as either (1) the ground truth differentially expressed features within a cluster or along a differentiation path from the Splatter single-cell RNA sequencing simulation study (see Splatter simulation) or (2) the features determined to be useful for classifying cells according to cell cycle phase using a random forest classifier trained on ground truth phase annotations from time-lapse imaging for the protein immunofluorescence imaging datasets (See Random forest, RPE analysis, PDAC analysis).

**Unsupervised clustering.** To evaluate feature selection method performance on retaining features that are informative for identifying canonical cell types, we performed unsupervised clustering on the data defined by the top $p$ ranked features from a feature selection strategy. More specifically, for each feature selection approach, clustering was performed on the selected data using the KMeans++ algorithm[105] with the number of centroids set as the same number of ground truth cell cycle phase labels for the protein immunofluorescence imaging datasets (RPE: $c = 4$, PDAC: $c = 5$).

To assess the accuracy of clustering assignments, we quantified a NMI score between the predicted cluster labels and the ground truth cell type labels. Normalized mutual information[166] is a clustering quality metric that measures the amount of shared information between two cell-to-cluster partitions (**u** and **v**, such that the $i$th entry $u_i$ gives the cluster assignment of cell $i$) as,

$$\mathrm{NMI} = \frac{2\mathrm{I}(\mathbf{u};\mathbf{v})}{\mathrm{H}(\mathbf{u})\mathrm{H}(\mathbf{v})}, \tag{25}$$

where, $\mathrm{I}(\mathbf{u};\mathbf{v})$ measures the mutual information between ground truth cell type labels **u** and cluster labels **v**, and $\mathrm{H}(\mathbf{u})$ or $\mathrm{H}(\mathbf{v})$ indicates the Shannon entropy or the amount of uncertainty for a given set of labels. Here, a score of 1 indicates that clustering on the selected features perfectly recovers the ground truth cell type labels. KMeans++ clustering was implemented using the KMeans function in scikit-learn v0.23.2.

**Protein-protein interaction networks.** In this work, we aimed to test whether features within DELVE dynamic clusters had experimental evidence of co-regulation as compared to random assignment. The STRING (search tool for the retrieval of interacting genes/proteins) database[122] is a relational database that computes protein association scores according to information derived from several evidence channels, including computational predictions (e.g., neighborhood, fusion, co-occurance), co-expression, experimental assays, pathway databases, and literature text mining. To assess the significance of protein interactions amongst features within a DELVE cluster, we performed a permutation test with a test statistic derived from STRING association scores using experimental evidence as follows.

Let $\mathcal{G}_p = (\mathcal{N}_p, \mathcal{E}_p)$ denote a graph of $p$ proteins from a DELVE cluster comprising the nodes $\mathcal{N}_p$, and $\mathcal{E}_p$ denote the set of edges, where edge weights encode the association scores of experimentally-derived protein-protein interaction evidence from the STRING database. Moreover, let $\mathcal{G}_r = (\mathcal{N}_r, \mathcal{E}_r)$ denote a graph of $r$ proteins randomly sampled without replacement from the full feature space $d$ such that $r = p$ comprising the nodes $\mathcal{N}_r$, and $\mathcal{E}_r$ denote the set of edges encoding the experimentally-derived association scores between those $r$ proteins from the STRING database. We compute the permutation $p$-value as described previously in ref. [167] as,

$$p\text{-value} = \frac{N+1}{R+1}. \tag{26}$$

Here $N$ indicates the number of times that $T_r \geq T_{\mathrm{obs}}$ out of $R$ random permutations ($R = 1000$), where $T_r$ is the average degree of a STRING association network from randomly permuted features as $T_r = {|\mathcal{N}_r|}/{|\mathcal{E}_r|}$, and $T_{\mathrm{obs}}$ is the average degree of a STRING association network from the features identified within a DELVE cluster as $T_{\mathrm{obs}} = {|\mathcal{N}_p|}/{|\mathcal{E}_p|}$. Of note, networks with higher degree are more connected, and thus show greater experimental evidence of protein-protein interactions. Experimental evidence-based association scores were obtained from the STRING database (stringdb) and networks were generated using networkx v3.1 package in python.

**Trajectory inference and analysis.** To evaluate how well feature selection methods can identify features that (1) recapitulate the underlying cellular trajectory and (2) can be used for trajectory analysis, we computed three metrics to assess trajectory preservation at different stages of inference: accuracy of the inferred cell-state trajectory graph, correlation of estimated pseudotime to the ground truth cell progression measurements, and significance and biological relevance of dynamic features identified following trajectory inference and regression analysis.

To obtain predicted cellular trajectories following feature selection, we performed trajectory inference using two approaches that were shown to outperform alternative methods for inference of simple or tree differentiation trajectories[23]. First, cellular trajectories were inferred using the diffusion pseudotime algorithm[31] based on 20 diffusion map components generated from a $k$-nearest neighbor graph ($k = 10$), where edge weights were determined by pairwise Euclidean distances between cells according to selected feature expression. For each feature selection approach, we estimated pseudotime using ten random root cells selected according to a priori biological knowledge: simulated datasets (cells with the smallest ground truth time annotation), 4i cell cycle datasets (cells with the youngest age from time-lapse imaging for the arrested (G0 phase) or proliferative (G1, S, G2, M phases) trajectories), CD8 differentiation dataset (cells from the d3 population), and DE differentiation dataset (cells from the d0 population). As a secondary approach, we inferred cellular trajectories using Slingshot[28]. Here, a minimum spanning tree was fit through cluster centroids defined according to a priori biological knowledge (e.g., cell type labels or time point labels), then pseudotime was estimated by projecting cells onto the principal curves fit through the PHATE embedding generated from selected feature expression (See PHATE visualizations). The root cluster for each dataset was defined as cluster with the smallest ground truth time annotation as introduced previously. Feature selection trajectory performance was subsequently assessed as follows.

1. Trajectory graph similarity: Partition-based graph abstraction (PAGA)[30] performs trajectory inference by constructing a coarse grained trajectory graph of the data. First cell populations are determined either through unsupervised clustering, graph partitioning, or a prior experimental annotations. Next, a statistical measure of edge connectivity is computed between cell populations to estimate the confidence of a cell population transition. To assess if feature selection methods retained features that represented coarse cell type transitions, we compared predicted PAGA trajectory graphs to ground truth cell

cycle reference trajectories curated from the literature[17]. First, PAGA connectivity was estimated between ground truth cell cycle phase groups using the $k$-nearest neighbor graph ($k = 10$) based on pairwise Euclidean distances between cells according to selected feature expression. We then computed the Jaccard distance between predicted and reference trajectories as,

$$d_j\left(\mathbf{W}_p, \mathbf{W}_r\right) = 1 - \frac{|\mathbf{W}_p \cap \mathbf{W}_r|}{|\mathbf{W}_p \cup \mathbf{W}_r|}. \tag{27}$$

$\mathbf{W}_p$ indicates the predicted cell type transition adjacency matrix, where each entry $W_{p,ij}$ represents the connectivity strength between cell populations $i$ and $j$ from PAGA and $\mathbf{W}_r$ indicates the reference trajectory adjacency matrix with entries encoding ground truth cell type transitions curated from the literature. Here, a lower Jaccard distance indicates that predicted trajectories better capture known cellular transitions.

2. Pseudotime correlation: To evaluate if feature selection methods retained features that accurately represent a cell's progression through a biological trajectory, we computed the Kendall rank correlation coefficient between estimated pseudotime following feature selection and ground truth cell progression annotations (e.g., the ground truth pseudotime labels generated from simulations, time-lapse imaging molecular age measurements).

3. Regression analysis: To identify genes associated with the inferred differentiation trajectory (e.g., CD8+ T cell, definitive endoderm differentiation trajectory) following feature selection, we performed regression analysis for each gene ($d = 500$) along estimated pseudotime using a negative binomial GAM. Genes were considered to be differentially expressed along the inferred lineage if they had a $q$ value $< 0.05$ following Benjamini-Hochberg false discovery rate correction[123].

4. Gene Ontology: To identify the biological relevance of differentially expressed genes associated with the different differentiation trajectories specific to each feature selection strategy, we performed gene set enrichment analysis on the set of significant genes from either highly variable gene, DELVE, Hotspot, Laplacian score, or RNA velocity feature selection using Enrichr[124]. Here, we considered the gene sets (e.g., mouse - CD8+ T cell differentiation, human - DE differentiation) from GO Biological Process 2023.

Diffusion pseudotime was implemented using the dpt function in Scanpy v1.9.3 in python, Slingshot was implemented using the slingshot v2.1.1 package in R v4.1.1, PAGA was implemented using the paga function in Scanpy v1.9.3 in python, GAM regression was implemented using the statsmodels v0.14.0 package in python, and gene set enrichment analysis was performed using the enrichr function in gseapy v1.0.4 package in python.

**PHATE visualizations.** To qualitatively compare lower dimensional representations from each feature selection strategy, we performed nonlinear dimensionality reduction using PHATE (potential of heat-diffusion for affinity-based transition embedding)[78] as this approach performs reasonably well for representing complex continuous biological trajectories. PHATE was implemented using the phate v1.0.11 package in python. Here, we used the same set of hyperparameters across all feature selection strategies (`knn = 30`, `t = 10`, `decay = 40`).

**Aggregate scores.** To rank feature selection methods on preserving biological trajectories in the presence of single-cell noise, we computed rank aggregate scores by taking the mean of scaled method scores across simulated single-cell RNA sequencing datasets from a trajectory type and noise condition (e.g., linear trajectory, dropout noise). More specifically, we first defined an overall method score per dataset as the median of each metric. Method scores were

subsequently min-max scaled to ensure datasets were equally weighted prior to computing the average.

## Reporting summary
Further information on research design is available in the Nature Portfolio Reporting Summary linked to this article.

## Data availability
The raw publicly available single-cell datasets used in this study are available in the Zenodo repository https://doi.org/10.5281/zenodo.4525425 for the RPE cell cycle dataset[168], the Zenodo repository https://doi.org/10.5281/zenodo.7860332 for the PDAC cell cycle datasets[164], and the Gene Expression Omnibus (GEO) under the accession code GSE131847 for the CD8+ T cell differentiation dataset[169]. All pre-processed datasets, including the DE differentiation dataset are available in the Zenodo repository https://doi.org/10.5281/zenodo.10534873[170]. The STRING database leveraged in this study is publicly available https://string-db.org/. The source data are provided with this paper and available in the Zenodo repository https://doi.org/10.5281/zenodo.10534873[170].

## Code availability
DELVE is implemented as an open-source Python package and is publicly available at https://github.com/jranek/delve. Source code including all functions for benchmarking feature selection methods including preprocessing, feature selection, evaluation, and plotting are publicly available at: https://github.com/jranek/delve_benchmark. Code is also publicly available in the Zenodo repository https://doi.org/10.5281/zenodo.10426508[171].

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

## Acknowledgements

We would like to thank Tarek Zikry, Garrett Sessions, and Alec Plotkin for their insightful discussions related to this work. This work was supported by the National Institutes of Health, F31-HL156433 (J.S.R.), 5T32-GM067553 (J.S.R.), R01-GM138834 (J.E.P.), The National Institutes of Allergy and Infectious Diseases of the National Institutes of Health under the award number 1R21AI171745-01A1 (N.S.), NSF CAREER Award 1845796 (J.E.P.), and NSF Award 2242980 (J.E.P.). Human H9 embryonic stem cells were obtained from WiCell and used with permission under WiCell Agreement No. 18-W0297 with the University of North Carolina at Chapel Hill. The funding mechanisms supporting the stem cell work (R01-GM138834) approved the use of the H9 stem cell line (0062) from the NIH registry.

## Author contributions

J.S.R. and W.S. conceptualized the study. J.S.R., N.S and J.E.P. designed the method and computational experiments. J.S.R. performed the data preprocessing, benchmarking, evaluation, and analysis. W.S. provided input for the RPE and PDAC cell cycle study. J.M. provided input for the CD8+ T cell study. M.R., S.C.W. and A.S.B. performed the DE differentiation experiments. J.S.R. wrote the manuscript with help from all authors. All authors read and approved of the final manuscript.

## Competing interests

The authors declare no competing interests.
