## [Peer Review File · Nature Communications]

DELVE: Feature selection for preserving biological trajectories in single-cell dataReviewer #1 (Remarks to the Author):

The authors introduced a method called DELVE (dynamic selection of locally covarying features), which is an unsupervised feature selection approach that aims to identify a representative subset of dynamically expressed molecular features. DELVE mitigates the effect of unwanted variation and models cell states from dynamic feature modules. The method was validated through simulations and experiments on single-cell RNA sequencing and immunofluorescence imaging data, showing improved characterization of cell populations and better recovery of cell type transitions. DELVE offers an alternative approach to improving trajectory inference and uncovering co-variation in biological trajectories. This research would be highly interesting and valuable for researchers in the relevant field. It has the potential to provide significant insights and contribute to the advancement of knowledge in the area of study.

Main concerns:

1) The authors conducted simulation studies using Splatter to generate 90 single-cell RNA sequencing datasets. Each dataset consisted of 1500 cells and 500 genes. However, to better mimic real single-cell studies, it would be beneficial for the authors to generate larger-scale datasets, such as including 10k or 20k genes in 10k cells.

The choice of 90 datasets was specifically designed to represent three trajectory structures, namely linear, bifurcation, and tree topologies. Each trajectory structure was represented by 30 datasets. They need to clearly state this when introducing the simulation experiments.

2) In the simulation studies, the authors utilized the Splatter simulator to generate datasets. However, it is recommended that they consider using a newer simulator called SymSim for their simulation studies.

3) They selected the top 100 ranked features for downstream analysis and evaluation. It would be valuable for them to explore the effects of selecting different numbers of top features, such as 50 or 200, in order to assess their impact on the results.

4) Fig 3 displays the overall ranked method performance of feature selection methods when subjected to different sources of single-cell noise (decreased signal-to-noise ratio, decreased library size, and increased dropout). However, it is necessary for the authors to provide further clarification regarding the specific degree of decrease or increase in these factors. Specifically, it would be helpful to specify the extent to which the signal-to-noise ratio is decreased, the magnitude of the decrease in library size, and the level of increased dropout.

5) Fig 4 d) shows the performance of the compared method using various evaluation metrics. However, it is suggested that the authors also include the evaluation metric RRMSE to assess the relative errors. This additional metric will provide a comprehensive evaluation of the methods' performance and allow for a more thorough comparison.

Minor:

In Fig 1, one of the outcomes is referred to as "dynamic feature importance". To accurately represent this outcome on the y-axis of the plot, it is recommended to label it as "gene importance" instead of simply "gene."

Reviewer #2 (Remarks to the Author):

The paper introduces an unsupervised feature selection method that aims to identify features that are discriminative of cell clusters or trajectories from the single-cell transcriptome data without relying on cell type or label information. Overall it is an interesting problem but the paper fell short in establishing the statistical rigor of the proposed method and the intuition behind its arbitrary choices.

1. Because it is not a model-based approach, my main concern is that the method does not model the distribution of the scRNA-seq data and therefore lacks rigorous measurement of statistical confidence like FDR or posterior probabilities in the feature selection.

2. Arbitrary choices such as number of neighbors per cell are required in step 1.

3. The intuition behind the core algorithm like equation 8 needs more work.
4. A lot of descriptions in the Results section (e.g., line 111-180) should have been under Methods.

Reviewer #3 (Remarks to the Author):

The authors developed a new feature extraction algorithm DELVE for trajectory reconstruction. Although large amount of comparison works have been done on simulated and experimental data to demonstrate the superiority of DELVE, I have a major concern. DELVE seems a combination of two existing methods hotspot and Laplacian score, the motivation of such combination is unclear, and DELVE failed to comprehensively benchmarked against the two seed methods. I think the authors need to provide more results and evidences to convince me, otherwise, DELVE is more like another method which may get some improvement on some metrics but unnecessary in practices. Other points are listed below.

1. The dynamic module of DELVE should be mentioned in section 'Overview of the DELVE algorithm'.
2. More trajectory estimation methods are required to evaluate the feature extraction methods, e.g., monocle2, monocle3, slingshot, RNA velocity.
3. In line 309, authors should state out of how many genes, the 13 cell cycle related genes were identified. Furthermore, gene set enrichment results of all identified genes should be provided to demonstrate cell cycle proliferation related terms are significant among all identified genes.
4. Fig 4b, 4d, the performance of DELVE was comparable to existing methods hotspot which compromise the motivation of a new algorithm. Furthermore, even though hotspot got slightly worse metric scores, it may not affect the results interpretation.
5. It is interesting to see the performance of hotspot in Fig 5, since hotspot performed highly comparable to DELVE on experimental data. To justify a new algorithm, the authors should demonstrate in which case the new algorithm works but not the existing ones. Furthermore, given the highly similar principles between DELVE and hotspot, such comparable performance are expected.
6. I notice hotspot and DELVE have similar performance on experimental data, but quite difference on simulate data. I doubt the how much the simulated data can mimic the real circumstances.
7. Line 459-466, authors did not show any aligned cell results, and the modeling building description is confusing. It is hard to image how to reconstruct the trajectory, what the trajectory looks likes, how to interpret results. Fig 5e did not show cell trajectory. The relationship between fig 5d and 5e is confusing. Since DELVE is an unsupervised method, what's the point to order cells on known markers, and what is the relationship of fig 5d to DEVLE. I would like to see the results of hotspot and Laplacian score on the CD8T data.
8. In line 25, authors called cell cycle as confounding source. However, in figure 5, DEVLE identify cell cycle signatures. The authors should demonstrate how DEVLE excludes confounding factors by keeping information of interest on simulated and experimental data.
9. In eq2, should Z' by Z ? It looks the feature extraction is on MSE. Please clarify it. In line 549, how to select the number of kmeans clusters? Eq 5 is not appropriate. A p-value should be calculated to identified significant features.
10. The DELVE seems just a combination of two existing methods, Laplacian score and hotspot. Therefore, the motivation of DELVE is unclear. It looks just another algorithm, which achieved similar results using similar methods. The author must show on experimental data what can DELVE do but not existing methods, especially hotspot and Laplacian score.
11. While following the reference code of "delve" on github (<https://github.com/jranek/delve>), I can't successfully create the conda environment of delve (CentOS Linux release 7.7.1908, python 3.9). Also, the code to produce the main results and figures should be posted. Please check the package code.

Reviewer #1 (Remarks to the Author):

The authors introduced a method called DELVE (dynamic selection of locally covarying features), which is an unsupervised feature selection approach that aims to identify a representative subset of dynamically expressed molecular features. DELVE mitigates the effect of unwanted variation and models cell states from dynamic feature modules. The method was validated through simulations and experiments on single-cell RNA sequencing and immunofluorescence imaging data, showing improved characterization of cell populations and better recovery of cell type transitions. DELVE offers an alternative approach to improving trajectory inference and uncovering co-variation in biological trajectories. This research would be highly interesting and valuable for researchers in the relevant field. It has the potential to provide significant insights and contribute to the advancement of knowledge in the area of study.

Main concerns:

1. The authors conducted simulation studies using Splatter to generate 90 single-cell RNA sequencing datasets. Each dataset consisted of 1500 cells and 500 genes. However, to better mimic real single-cell studies, it would be beneficial for the authors to generate larger-scale datasets, such as including 10k or 20k genes in 10k cells.

We thank the reviewer for this suggestion. Based on the reviewer's recommendation in comment 2, we have additionally benchmarked feature selection methods on larger scale simulated single-cell RNA sequencing datasets containing 10k cells and 20k genes using SymSim (Zhang et. al. PMID 31197158), as this approach has been shown in a recent benchmarking study (Cao et. al. PMID 34824223) to be amongst the top ranking methods for reasonably simulating single-cell RNA sequencing data as measured by the accuracy in estimating data properties (e.g. library size, TMM, mean expression, scaled variance, fractions of zeros, cell and gene correlation, etc.) and the ability to preserve biological signals (e.g. differentially expressed genes, differentially variable genes, etc.).

In this simulation study, we simulated five tree differentiation trajectories containing 10,000 cells, 20,000 genes, and 4 cell types by modifying the mean mRNA capture efficiency rate in SymSim. Here, we aimed to evaluate how well feature selection methods could identify genes that define cellular trajectories when subjected to a reduction in the total mRNA count. We assessed feature selection method performance according to (1) the preservation of cell types (as measured by kNN classification accuracy), and (2) the preservation of cellular ordering (as measured by Spearman rank correlation between estimated pseudotime using diffusion pseudotime (Haghverdi et. al. PMID 27571553) or Slingshot (Street et. al. PMID 29914354) to the ground truth simulated measurements). For more details on the SymSim simulation design, see lines 946-965 under the SymSim subsection the Methods section. Similar to the Splatter simulation study, overall we found that unsupervised similarity-based feature selection methods, DELVE, Laplacian score, and Hotspot outperformed the alternative feature selection methods and achieved higher cell type classification accuracy scores and

higher pseudotime correlations (See Supplementary Figure 13).

Moreover, as recommended by Reviewer 1 in Comment 3, we have used these data to additionally evaluate the effect of selecting different numbers of features on trajectory preservation for twelve different feature selection methods (See Supplementary Figure 13). We observed that variance-based approaches (e.g. max variance, highly variable gene selection) required more features (e.g. 2000 genes) to obtain similar cell type classification accuracy and pseudotime correlation scores. In contrast, DELVE, Laplacian score, and Hotspot achieved higher scores with smaller representative subsets of features (e.g. 100 genes) and were more robust to the number of selected genes (See Supplementary Figure 13). Furthermore, neighborhood variance and SCMER identified the smallest number of genes (e.g. 50 genes) and they were often biologically predictive.

Lastly, we have used these simulated datasets to further evaluate DELVE's sensitivity to parameter choices (See lines 783-795 under the 'DELVE's sensitivity to parameter choices' subsection in the Methods section and Supplementary Figure 14). Overall, we found that DELVE was robust to the number of nearest neighbors, the number of subsampled cells, and the number of clusters in dynamic module identification, as measured by unsupervised clustering or trajectory inference performance following feature selection (See Supplementary Figures 14, 15).

The choice of 90 datasets was specifically designed to represent three trajectory structures, namely linear, bifurcation, and tree topologies. Each trajectory structure was represented by 30 datasets. They need to clearly state this when introducing the simulation experiments.

We thank the reviewer for this comment and have clarified the splatter simulation design on lines 167-170 in the 'Simulation study 1: simulating single-cell RNA sequencing data with Splatter' in the Main text, as well as in line 909 in the 'Splatter simulation' subsection in the Methods section.

2. In the simulation studies, the authors utilized the Splatter simulator to generate datasets. However, it is recommended that they consider using a newer simulator called SymSim for their simulation studies.

Based on the reviewer's recommendation, we have additionally benchmarked feature selection methods using a secondary simulation approach, SymSim (Zhang et. al. PMID 31197158) (See Reviewer 1 Comment 1, Supplementary Figure 13). For more details on the simulation design, see lines 295-319 under the 'Simulation study 2: simulating single-cell RNA sequencing data with SymSim' subsection in the Main text and in lines 946-965 under the SymSim subsection in the Methods section.

3. They selected the top 100 ranked features for downstream analysis and evaluation. It would be valuable for them to explore the effects of selecting different numbers of top features, such as 50 or 200, in order to assess their impact on the results.

We agree with the reviewer's suggestion and have additionally evaluated feature selection method performance by varying the number of selected features (50, 100, 250, 500, 1000, 2000 genes) on SymSim simulated single-cell RNA sequencing data (See Reviewer 1 Comment 1, lines 946-965 under the SymSim subsection the Methods section, Supplementary Figure 13).

4. Fig 3 displays the overall ranked method performance of feature selection methods when subjected to different sources of single-cell noise (decreased signal-to-noise ratio, decreased library size, and increased dropout). However, it is necessary for the authors to provide further clarification regarding the specific degree of decrease or increase in these factors. Specifically, it would be helpful to specify the extent to which the signal-to-noise ratio is decreased, the magnitude of the decrease in library size, and the level of increased dropout.

We agree with the reviewer's suggestion and have provided further context regarding the degree to which noise was introduced in the Splatter simulation study. Specifically, we have quantified the average coefficient of variation amongst genes following a decrease in the signal-to-noise ratio (See lines 931-933), the average library size following reduction in the total mRNA count (See lines 939-940), as well as the average undersampling percentage to indicate the level of increased dropout (See lines 942-943 in the 'Splatter simulation' subsection in the Methods section).

5. Fig 4 d) shows the performance of the compared method using various evaluation metrics. However, it is suggested that the authors also include the evaluation metric RMSE to assess the relative errors. This additional metric will provide a comprehensive evaluation of the methods' performance and allow for a more thorough comparison.

Based on the reviewer's suggestion, we have evaluated the RMSE between true age and predicted age using a Random Forest regression model trained on the selected features from each feature selection strategy and the ground truth age measurements from time-lapse imaging. Furthermore, we replaced the MSE panel in Figure 4d with RMSE to provide a more intuitive explanation of method performance with respect to relative age prediction errors in hours (See Figure 4d, lines 425-428 in the Main text, and line 1100 in the Methods section).

Minor:

1. In Fig 1, one of the outcomes is referred to as "dynamic feature importance". To accurately represent this outcome on the y-axis of the plot, it is recommended to label it as "gene importance" instead of simply "gene."

We thank the reviewer for this comment and agree that labeling the y-axis as 'gene' is misleading. We have modified the y-axis on this panel to be 'feature expression' vs. 'gene' as we think this more accurately depicts the panel within the plot (See Figure 1).

Reviewer #2 (Remarks to the Author):

The paper introduces an unsupervised feature selection method that aims to identify features that are discriminative of cell clusters or trajectories from the single-cell transcriptome data without relying on cell type or label information. Overall it is an interesting problem but the paper fell short in establishing the statistical rigor of the proposed method and the intuition behind its arbitrary choices.

1. Because it is not a model-based approach, my main concern is that the method does not model the distribution of the scRNA-seq data and therefore lacks rigorous measurement of statistical confidence like FDR or posterior probabilities in the feature selection.

While we agree with the reviewer that a model-based approach would allow us to more rigorously evaluate the statistical significance of ranked features, we note that many commonly used feature selection approaches are model free (Breiman Mach. Learn. 2001, Satija et. al. PMID 25867923, He et. al. Neurips 2005, Liang et. al. PMID 36969355, Cai et. al. ACM SIGKDD 2010, Welch et. al. PMID 27215581) and that it is still possible for users to empirically derive a measure of statistical confidence by generating a distribution of accuracy metrics across many runs of the method. We have clarified this point on lines 140-141 in the Main text.

In this study, we aimed to rigorously evaluate our feature selection method on selecting features that recapitulate biological trajectories by benchmarking our approach to both model-based (hotspot) and data driven approaches (e.g. Random Forest classification, neighborhood variance, Laplacian score, SCMER, MCFS, highly variable gene selection, max variance) on datasets that had ground truth annotations. Moreover, for each evaluation metric, we reported the results across 20 random initializations of each algorithm to demonstrate the reproducibility of selected features and downstream performance.

2. Arbitrary choices such as number of neighbors per cell are required in step 1.

We thank the reviewer for this comment and completely agree that providing intuition behind choices in parameters is crucial and should be described in more detail. We have provided further context for parameter selection on lines 749-782 in the 'Guidelines on parameter selection' subsection in the Methods section.

Moreover, we have evaluated DELVE's sensitivity to choices in hyperparameters by varying the number of neighbors per cell, the number of subsampled cells, and the

number of clusters using both simulated single-cell RNA sequencing data (See Supplementary Figure 14) and real proteomic imaging data (See Supplementary Figure 15). Overall, we found DELVE feature selection to be robust across a range of parameter choices (as measured by unsupervised cell population identification or trajectory inference performance), with the most sensitive parameter often being the number of clusters in KMeans++ clustering when identifying dynamic modules (See Supplementary Figures 14-15). For more details on the evaluation (See 'DELVE's sensitivity to parameter choices' subsection in the Methods section on lines 783-795).

3. The intuition behind the core algorithm like equation 8 needs more work.

We thank the reviewer for this comment and have (1) further elaborated on the underlying assumptions and intuition behind graph signal processing and neighborhood-based variations in feature values (Dong et. al. doi 10.1109/msp.2020.3014591, Shuman et. al. doi 10.1109/msp.2012.2235192), (2) provided additional references for the Laplacian quadratic form and smoothness in equation 8, (3) further described how noisy features can impact the cell similarity graph, the graph Laplacian matrix, and feature ranking (Charrouf et. al. PMID 33575604, Lindenbaum et. al. Neurips 2021, Shaham et. al. PMID 35500458, and (4) further described the intuition behind using the Laplacian quadratic form on a cell similarity graph defined by dynamically expressed molecular features that constitute core regulatory complexes in order to rank a feature's association with the underlying cellular trajectory (See lines 700-748 under the 'DELVE' subsection in the Methods section).

4. A lot of descriptions in the Results section (e.g., line 111-180) should have been under Methods.

We thank the reviewer for this suggestion and will consult the editor about moving this section from the Results to the Methods.

Reviewer #3 (Remarks to the Author):

The authors developed a new feature extraction algorithm DELVE for trajectory reconstruction. Although a large amount of comparison works have been done on simulated and experimental data to demonstrate the superiority of DELVE, I have a major concern. DELVE seems a combination of two existing methods hotspot and Laplacian score, the motivation of such combination is unclear, and DELVE failed to comprehensively benchmarked against the two seed methods. I think the authors need to provide more results and evidences to convince me, otherwise, DELVE is more like another method which may get some improvement on some metrics but unnecessary in practice. Other points are listed below.

1. The dynamic module of DELVE should be mentioned in section 'Overview of the DELVE algorithm'.

We thank the reviewer for this comment and have further clarified the terminology within the dynamic module identification step in the 'Overview of DELVE algorithm' section on lines 73-100. We have additionally provided references to the appropriate methods subsections to give further context on dynamic module selection and feature ranking.

2. More trajectory estimation methods are required to evaluate the feature extraction methods, e.g., monocle2, monocle3, slingshot, RNA velocity.

We thank the reviewer for these recommendations and completely agree that more trajectory inference methods are required to fully evaluate feature selection performance. In addition to diffusion pseudotime and PAGA, we have additionally inferred cellular trajectories using Slingshot (Street et al., PMID 29914354) following feature selection on both simulated (See Supplementary Figure 13) and real single-cell datasets (See Supplementary Figures 28, 30). Overall, we found that feature selection trajectory preservation results were concordant when inferring cellular trajectories with diffusion pseudotime and Slingshot. We note that for the SymSim simulation study, Slingshot often achieved higher pseudotime correlations to the ground truth measurements and obtained more stable results for all feature selection approaches when initialized with ground truth cell type labels, as directly compared to the more unsupervised diffusion pseudotime approach (See Supplementary Figure 13b,d). For more details on implementation and evaluation, see 'Trajectory inference and analysis' subsection within the Methods Section on lines 1146-1199.

Moreover, we have evaluated RNA velocity's ranked likelihood genes (Bergen et. al., PMID 32747759) on preserving cellular trajectories using an additional real single-cell RNA sequencing differentiation dataset that contained unspliced and spliced molecular counts (See lines 532-597 under the subsection 'Characterizing human embryonic stem cell differentiation into the definitive endoderm' the Main text, lines 1065-1072 under the 'DE differentiation analysis' subsection in the Methods, Figure 6e-h, and Supplementary Figures 29-30). Strikingly, we found that DELVE identified ~30 percent more lineage-specific transcription factors than HVG and RNA velocity (See Figure 6g). Moreover, we found that RNA velocity often failed to appropriately model and identify key transcription factors driving definitive endoderm differentiation, as they exhibited more switch-like or transient kinetic behavior (See Figure h). In contrast, DELVE was able to successfully identify these dynamic transcription factors involved the (1) core pluripotency network (SOX2, POU5F1) (2) organization and formation of the primitive streak and mesendoderm (CDX1, CDX2, EOMES, GSC, OTX2, MESP1, MESP2, LHX1), and (3) known regulators involved in definitive endoderm cell fate specification (SOX17, FOXA2).

3. In line 309, authors should state out of how many genes, the 13 cell cycle related genes were identified. Furthermore, gene set enrichment results of all identified genes should be provided to demonstrate cell cycle proliferation related terms are significant among all

identified genes.

We thank the reviewer for this comment and suggestion for clarification. We have further clarified that the 13 cell cycle proteomic imaging features were identified from the entire dataset consisting of 241 imaging features, and that cell cycle proliferation related terms are indeed significant amongst all profiled imaging features (as sought out and designed for by the initial experiment) (See lines 339-350 in the 'Revealing molecular trajectories of proliferation and cell cycle arrest' subsection in the main text). It should be further noted that this study aims to identify proteomic imaging features related to cell cycle progression (e.g. changes in morphology, expression and localization of specific cell cycle markers) amongst all extracted imaging-derived features. Although all profiled proteins are indeed cell cycle-specific, we note that many are not expressed within all regions of the cell leading to increasing amounts of noisy imaging-derived features. This propagates into downstream analysis and hinders feature selection performance, clustering/ grouping of cells according to cell cycle phases, and trajectory inference performance. This study demonstrates how DELVE can be used to identify proteomic imaging-derived features that are strongly associated with the cell cycle by capturing changes in cell morphology and protein localization from a feature list that contains all extracted imaging features (noisy or otherwise). In doing so, DELVE feature selection results in improved identification and ordering of the progression of proliferation and arrest cell states as quantified by time-lapse imaging annotations (Figure 4, Supplementary Figures 19-27). This is in comparison to the alternative methods that are often overwhelmed by noisy features (e.g. extracted imaging-derived measurements that contain low signal-to-noise ratio).

4. Fig 4b, 4d, the performance of DELVE was comparable to existing methods hotspot which compromise the motivation of a new algorithm. Furthermore, even though hotspot got slightly worse metric scores, it may not affect the results interpretation.

While we agree with the reviewer that slight differences in metric scores might not affect the interpretation of the results (e.g. Figure 4 when comparing DELVE to hotspot on classification accuracy), we note that there are quite a few examples where DELVE significantly outperforms the alternative methods that would lead to different interpretations of results. This is evident by cell state quantification (e.g. classification performance, clustering performance, precision of phase-specific features), cell transition preservation (e.g. correlation of estimated cellular ordering to the ground truth measurements, trajectory graph estimation), as well as considerable differences in low dimensional projections as shown below (e.g. DELVE vs. Laplacian score in Figure 4, DELVE vs. hotspot/Laplacian score/alternative methods in Supplementary Figures 19, 20, 22, 27).

Figure 4: DELVE (black) vs. Laplacian score (red) in inferring RPE cell cycle

Supplementary Figure 19: DELVE (black) vs. hotspot and Laplacian score (red) in inferring BxPC3 PDAC cell cycle

Supplementary Figure 20: DELVE (black) vs. hotspot and Laplacian score (red) in inferring CFPAC PDAC cell cycle

b

c

Supplementary Figure 22: DELVE (black) vs. hotspot and Laplacian score (red) in inferring MiaPaCa PDAC cell cycle

b

c

Supplementary Figure 27: DELVE (black) vs. hotspot and Laplacian score (red) in inferring UM53 PDAC cell cycle

5. It is interesting to see the performance of hotspot in Fig 5, since hotspot performed highly comparable to DELVE on experimental data. To justify a new algorithm, the authors should demonstrate in which case the new algorithm works but not the existing ones. Furthermore, given the highly similar principles between DELVE and hotspot, such comparable performance are expected.

We thank the reviewer for this comment and have additionally evaluated Hotspot and the Laplacian score on the CD8 T cell single-cell RNA sequencing dataset (See Figure 5, Supplementary Figure 28). Moreover, to provide further context and evaluation of feature selection methods on identifying features that preserve biological trajectories from real single-cell RNA sequencing data, we have additionally evaluated DELVE on another real single-cell dataset, where we used multiplexed single-cell RNA sequencing to profile human embryonic stem cells differentiating to the definitive endoderm (See Supplementary Figures 29-30). Overall, we found that similarity-based feature selection approaches like DELVE, Hotspot, and the Laplacian score facilitated increased performance on inferring cellular trajectories from single-cell RNA sequencing data, as compared to the alternative variance-based approaches (e.g. highly variable gene selection). However, we highlight that DELVE outperforms Hotspot and the Laplacian score on inferring cellular trajectories from noisy proteomic imaging data as described previously in Reviewer 3, Comment 4.

While we agree with the reviewer that the general principle of identifying modules and performing feature selection using a cell similarity graph is similar between hotspot and DELVE, we note that the methods differ in their design, the way in which cell similarity is defined when performing feature selection, its computation of modules, and its application. We have further clarified these differences in Supplementary Table 1. Hotspot performs feature selection by evaluating a gene's association with a similarity graph defined by all profiled features using a test statistic based on local autocorrelation. Following feature selection, modules are identified by grouping features according to local co-expression patterns. In contrast to hotspot and the alternative feature selection methods, DELVE uses a bottom-up approach and computes pairwise changes in expression of features along cell states to redefine cell similarity according to a subset of dynamically expressed features. All features are then ranked according to their association with an approximate cell trajectory graph for trajectory-preserving feature selection. By excluding noisy features when defining the cell similarity graph for feature ranking, DELVE mitigates the effect of unwanted sources of variation confounding feature selection performance.

The differences in design between DELVE, Hotspot, and the Laplacian score can be shown in the improved performance of DELVE on inferring cell cycle trajectories using 4i, especially from the more challenging noisy real experimental PDAC 4i cell cycle datasets (Supplementary Figures 19, 20, 22, 27).

6. I notice hotspot and DELVE have similar performance on experimental data, but quite difference on simulate data. I doubt the how much the simulated data can mimic the real circumstances.

We thank the reviewer for this comment. For this simulation study, we evaluated feature selection methods on 90 datasets that represented three trajectory structures (linear, bifurcation, and tree topologies). Here, each trajectory structure was represented by 30 datasets, where parameters were estimated from a real single-cell RNA sequencing dataset consisting of mesoderm progenitors (Loh et al., PMID 27419872) and increasing amounts of biological/technical noise was added (e.g. signal to noise, library size, dropout). We argue that this simulation design represents a robust comparison of feature selection methods on inferring biological trajectories from single-cell RNA sequencing data as the parameters were estimated from real data, and we can modulate the trajectory structure and the amount of noise, while having ground truth information for quantifying the performance. This is in direct comparison to evaluating feature selection method performance on one real single-cell RNA sequencing dataset that may only represent one particular type of technical challenge and lack of ground truth for evaluation.

We agree with the reviewer that existing single-cell RNA sequencing simulation software has the potential to generate parameters or distributions of counts that can fail to capture biologically-relevant data or the extent of technical limitations (e.g. efficiency of mRNA capture). Therefore, to more comprehensively evaluate the performance of feature selection methods on single-cell RNA sequencing data, we've additionally benchmarked feature selection methods using a secondary simulation approach SymSim (Zhang et. al., PMID 31197158) as this approach has been shown in a recent benchmarking study (Cao et. al. PMID 34824223) to be amongst the top ranking methods for reasonably simulating single-cell RNA sequencing data as measured by the accuracy in estimating data properties (e.g. library size, TMM, mean expression, scaled variance, fractions of zeros, cell and gene correlation, etc.) and the ability to preserve biological signals (e.g. differentially expressed genes, differentially variable genes, etc.).

As mentioned previously to Reviewer 1 in Comment 1, in this additional simulation study, we simulated five tree differentiation trajectories containing 10,000 cells, 20,000 genes, and 4 cell types, by modifying the mean mRNA capture efficiency rate in SymSim. Here, we aimed to evaluate how well feature selection methods could identify genes that define cellular trajectories when subjected to a reduction in the total mRNA count. We assessed feature selection method performance according to (1) the preservation of cell types (as measured by kNN classification accuracy), and (2) the preservation of cellular ordering (as measured by Spearman rank correlation between estimated pseudotime using diffusion pseudotime (Haghverdi et. al. PMID 27571553) or Slingshot (Street et. al. PMID 29914354) to the ground truth simulated measurements). For more details on the SymSim simulation design, see lines 946-965 under the SymSim subsection the Methods section. Similar to the Splatter simulation study, overall we found that

unsupervised similarity-based feature selection methods, DELVE, Laplacian score, and Hotspot outperformed the alternative feature selection methods on trajectory preservation and were more robust to the number of selected genes (See lines 295-319, Supplementary Figure 13). Overall, these results suggest that unsupervised similarity-based feature selection methods outperform variance-based approaches on preserving cellular trajectories when evaluated on simulated single-cell RNA sequencing data.

While we agree with the reviewer that the performance of DELVE, Hotspot, and the Laplacian score are comparable on some single-cell RNA sequencing datasets (e.g. Figure 5, Supplementary Figure 30), we note that DELVE outperforms the alternative methods on many experimental 4i datasets (DELVE vs. Laplacian score in Figure 4 and Supplementary Figures 19-27) and (DELVE vs. hotspot in Figure 4 and Supplementary Figures 19, 20, 22, 27), which is likely due to different data modalities (e.g. 4i, single-cell RNA sequencing) which have inherently different technical challenges (e.g. signal to noise, dropout).

7. Line 459-466, authors did not show any aligned cell results, and the modeling building description is confusing. It is hard to image how to reconstruct the trajectory, what the trajectory looks like, how to interpret results. Fig 5e did not show cell trajectory. The relationship between fig 5d and 5e is confusing. Since DELVE is an unsupervised method, what's the point to order cells on known markers, and what is the relationship of fig 5d to DELVE. I would like to see the results of hotspot and Laplacian score on the CD8T data.

We thank the reviewer for this comment and have additionally evaluated Hotspot and the Laplacian score on the CD8 T cell dataset (See Figure 5, Supplementary Figure 28) as well as another real single-cell RNA sequencing dataset containing human embryonic stem cells differentiating into the definitive endoderm (See Figure 6, Supplementary Figures 29-30) to provide a comparison of feature selection methods on real single-cell RNA sequencing data. Moreover, we have simplified and clarified the CD8 T cell differentiation study design on lines 511-531 in the 'Identifying molecular drivers of CD8+ T cell effector and memory formation' subsection in the main text, as well as provided lower dimensional visualizations of the cellular trajectories colored according to (1) sampling day and (2) estimated pseudotime for both datasets and all feature selection methods.

In this study, we aimed to evaluate feature selection method performance on identifying features that define (1) the memory T cell lineage (See Figure 5, Supplementary Figure 28) or (2) the definitive endoderm lineage (See Figure 6, Supplementary Figures 29-30). We first performed unsupervised feature selection to identify the top 500 ranked features according to a feature selection strategy (e.g. DELVE, HVG, hotspot, Laplacian score, RNA velocity). Trajectory inference was then performed using either diffusion pseudotime or slingshot according to the expression of those selected features. To

identify the genes that significantly changed along the inferred cellular trajectory, we fit a generalized additive model with estimated pseudotime as the covariate. Gene set enrichment analysis was then performed on those significant dynamic features to identify pathway-related terms. Similar to the simulated single-cell RNA sequencing studies, we found that similarity-based feature selection methods (e.g. DELVE, Hotspot, and the Laplacian score) identified more lineage-specific genes as compared to both variance-based feature selection (HVG) and RNA velocity ranked likelihood genes (See Figure 5d-f, Figure 6e-h, Supplementary Figure 28-30).

8. In line 25, authors called cell cycle as confounding source. However, in figure 5, DELVE identify cell cycle signatures. The authors should demonstrate how DELVE excludes confounding factors by keeping information of interest on simulated and experimental data.

Based on the reviewer's suggestion, we have further elaborated on how DELVE can be used to exclude confounding factors, while retaining information of interest in the Discussion section on lines 641-643. In this study, we have extensively demonstrated how DELVE can be used in an unsupervised framework to identify dynamic modules of interest and exclude noisy features that might be confounding trajectory inference (e.g. Figures 2-6, Supplementary Figures 1-9, 13, 17, 19-30). For additional functionality, we have added an extension of DELVE to allow the user to include or exclude dynamic modules of interest prior to ranking feature importance. By performing pathway analysis on each dynamic module, the user can decidedly include or exclude dynamic modules (e.g. cell cycle modules) according to those pathways of interest if they so choose.

9. In eq2, should \tilde{Z} by Z ? It looks the feature extraction is on MSE. Please clarify it. In line 549, how to select the number of kmeans clusters? Eq 5 is not appropriate. A p-value should be calculated to identified significant features.

We would like to thank the reviewer for these comments for clarification. To clarify equation 2, for each neighborhood, DELVE computes the average pairwise change or difference in the expression of features across subsampled neighborhoods defined by the initial kNN graph. The dimensions of this matrix are [number neighborhoods x number of features]. Here, each row of the matrix represents a distinct neighborhood and the values correspond to the average pairwise change in expression for that neighborhood across all features. This is not equivalent to mean squared error, as mean squared error does not compute pairwise differences, and the differences in equation 2 are not squared. We have clarified this equation in the Identification of Modules subsection in the Methods section on lines 681-683.

Moreover, we have provided further context for parameter selection, including the number of kmeans clusters on lines 749-782 in the 'Guidelines on parameter selection' subsection in the Methods section. Furthermore, we have evaluated DELVE's sensitivity to choices in hyperparameters by varying the number of neighbors per cell, the number

of subsampled cells, and the number of clusters using both simulated single-cell RNA sequencing data (See Supplementary Figure 14) and real proteomic imaging data (See Supplementary Figure 15). Overall, we found DELVE feature selection to be robust across a range of parameter choices, as measured by unsupervised clustering and trajectory inference performance. For more details on the evaluation, See 'DELVE's sensitivity to parameter choices' subsection in the Methods section on lines 783-795.

We agree with the reviewer that a permutation p-value should be calculated to identify significant features and that the terminology used was misleading. In this study, we followed a similar approach to Ref. (Welch et. al., PMID 27215581) and defined dynamic modules of features as those that had a sample variance greater than random selection. We have removed the 'significance' terminology and have modified these subsections accordingly on lines 687-689.

10. The DELVE seems just a combination of two existing methods, Laplacian score and hotspot. Therefore, the motivation of DELVE is unclear. It looks just another algorithm, which achieved similar results using similar methods. The author must show on experimental data what can DELVE do but not existing methods, especially hotspot and Laplacian score.

We thank the reviewer for this comment and have more clearly articulated feature selection method performance with respect to data modalities and technical challenges in the Discussion section on lines 607-635 to provide practical guidance on when one method should be used over another. As mentioned in Reviewer 3 comment 5, we note that DELVE differs from Hotspot and the Laplacian score in its bottom-up framework and selection of dynamically expressed features when defining the cell similarity graph for ranking feature importance (See Supplementary Table 1). The motivation behind excluding noisy or irrelevant features prior to defining the cell similarity graph and ranking feature importance is in the ability for noisy features to overwhelm true biological differences, where the variability in noisy features can outweigh and mask the variability of informative ones (Charroux et. al. PMID 33575604, Lindenbaum et. al. Neurips 2021, Shaham et. al. PMID 35500458). This impacts feature selection and trajectory inference performance.

To further illustrate the differences in performance between DELVE, Hotspot, and the Laplacian score in more detail and showcase the robustness of our bottom-up approach, we evaluated the ability for feature selection methods to recover cell cycle defining features from immunofluorescence imaging data with different amounts of noisy variables (See Supplementary Figure 17, also shown below). To do so, we generated multiple RPE immunofluorescence imaging datasets, where each dataset was initialized with the same set of ground truth phase-specific features (i.e., the top 30 predictive features of phase by training a random forest classifier on phase labels from time-lapse imaging). We then added a fixed amount of noisy variables ($j = 100$ to 500) to each dataset by randomly sampling features from the experimental RPE dataset with

replacement. For each dataset and feature selection method, we selected the top 30 ranked features and compared the performance of DELVE, Hotspot, and the Laplacian score on various trajectory tasks. Overall, DELVE achieved the highest recovery of phase-specific features, cell population recovery (NMI between cluster labels and ground truth phase annotations), and trajectory recovery (highest correlation between estimated pseudotime and ground truth age annotations). In contrast, as the amount of noisy variables increased, Hotspot and the Laplacian score identified less phase-specific features and more noisy features. The cell similarity graph was more densely connected and less clusterable (higher clustering coefficient, increasing path length) which resulted in worse cell population and trajectory recovery. Results are shown over 5 random trials.

These results demonstrate the motivation and necessity of DELVE's bottom-up approach, and highlight how DELVE's different algorithmic design can be used to consistently recover cellular trajectories. In contrast, the alternative approaches (e.g., Hotspot and the Laplacian score) are often overwhelmed by noisy features that impact their overall precision and trajectory recovery.

Moreover, we further note the this improved performance and robustness of DELVE is demonstrated on many real experimental datasets that contain irrelevant and noisy

extracted proteomic imaging-derived features (DELVE vs. Laplacian score in Figure 4 and Supplementary Figures 19-27) and (DELVE vs. hotspot in Figure 4 and Supplementary Figures 19, 20, 22, 27).

11. While following the reference code of "delve" on github (<https://github.com/jranek/delve>), I can't successfully create the conda environment of delve (CentOS Linux release 7.7.1908, python 3.9). Also, the code to produce the main results and figures should be posted. Please check the package code.

We thank the reviewer for trying to test out the reference code of DELVE on the GitHub repository (<https://github.com/jranek/delve>) and apologize for the difficulty in creating the conda environment on Linux. To aid in installation of the necessary dependencies, we have published DELVE as a python package on pypi for installation with pip (<https://pypi.org/project/delve-fs/>). Moreover, as two alternatives, we have modified the conda environment yml file to be os independent and have listed all versions of the software used for manual installation if needed. As mentioned in the manuscript, we have additionally provided an alternative GitHub repository (https://github.com/jranek/delve_benchmark) for benchmarking feature selection methods. This repository contains all of the functions necessary for reproducing the main results in the paper, as well as the figures. To further aid in transparency, we will add jupyter notebooks to this GitHub repository for reproducing the figures.

Reviewer #1 (Remarks to the Author):

The authors have fully addressed my comments.
Thanks!

Reviewer #2 (Remarks to the Author):

All my comments are addressed.

Reviewer #3 (Remarks to the Author):

Generally, I did not see the answers of my key questions: what can DELVE bring to us but not hotspot. Please pinpoint at least one case directly.

1. In replying my previous points 4, although the authors demonstrate the dynamic features associated with different genes, and DELVE achieved improvement in some scenarios, it still not showed a consistent improvement. I suggest the author illustrate the average performance of various method across scenarios, e.g., average arrest pseudotime corr, et al.
2. For my previous point 5, I have double checked the Fig.5, S29, S30, and cannot conclude DELVE is superior than hotspot. From Fig 5f, hotspot seems even better as it can identify more relevant gene terms. In addition, the authors failed to appropriately response my comment: in which case the new algorithm works but not the existing ones. Generally, DELVE performs similar as hotspot. The authors explained DELVE have different design from hotspot. Although I can take this, but to publish in a journal like NC, it is important to demonstrate how the work can bridge a gap.
3. For Eq 5, a p-value is still helpful.

Reviewer #1 (Remarks to the Author):

The authors have fully addressed my comments. Thanks!

Reviewer #2 (Remarks to the Author):

All my comments are addressed.

Reviewer #3 (Remarks to the Author):

Generally, I did not see the answers of my key questions: what can DELVE bring to us but not hotspot. Please pinpoint at least one case directly.

1. In replying my previous points 4, although the authors demonstrate the dynamic features associated with different genes, and DELVE achieved improvement in some scenarios, it still not showed a consistent improvement. I suggest the author illustrate the average performance of various method across scenarios, e.g., average arrest pseudotime corr, et al.

We thank the reviewer for this suggestion and have computed the average performance of feature selection methods across ten 4i datasets (See lines 464-467, Supplementary Figure 28 as shown below where * indicates the method with the highest average across 4i datasets). DELVE achieves consistent improvement over the alternative unsupervised feature selection methods, including Hotspot and the Laplacian Score, across all tasks. Most notably, DELVE achieves higher precision of cell cycle specific features and improved inference of cell cycle trajectories.

- For my previous point 5, I have double checked the Fig.5, S29, S30, and cannot conclude DELVE is superior than hotspot. From Fig 5f, hotspot seems even better as it can identify more relevant gene terms. In addition, the authors failed to appropriately response my comment: in which case the new algorithm works but not the existing ones. Generally, DELVE performs similar as hotspot. The authors explained DELVE have different design from hotspot. Although I can take this, but to publish in a journal like NC, it is important to demonstrate how the work can bridge a gap.

In this study, we demonstrated that DELVE consistently outperforms the alternative feature selection methods on identifying trajectory-specific features from 4i data (See Supplementary Figure 28), and the response to question 1. While alternative similarity-based feature selection methods have the potential to identify smoothly varying features, they rely on a cell similarity graph defined by all features and can fail to identify relevant features when the number of noisy features far outweigh the number of informative ones. We demonstrated this in Supplementary Figure 17 as shown below, where we evaluated the impact of noisy features on feature selection method performance.

DELVE consistently achieved the highest recovery of phase-specific features, NMI score measuring cell population recovery, and highest correlation between estimated pseudotime and the ground truth age measurements from time lapse imaging. In contrast, as the number of noisy features increased, Hotspot and the Laplacian Score selected less phase-specific features and more noisy features, resulting in worse precision and trajectory inference performance. DELVE is able to mitigate this limitation

by excluding noisy features prior to feature ranking, which is crucial for trajectory analysis in imaging data, as they often contain hundreds or thousands of noisy extracted imaging measurements. A specific example of this limitation confounding feature selection approaches and trajectory inference is in Supplementary Figure 27, where DELVE is the only approach that can recover cell cycle trajectories.

Additional examples are shown in Supplementary Figures 19, 20, 22. These results demonstrate the necessity of using a bottom-up approach when performing feature selection from noisy proteomic imaging data and showcases the robustness of DELVE on creating a representation of the data that is faithful to underlying cellular trajectory structure across noisy datasets.

Moreover, this work also addresses a critical gap in the literature by benchmarking twelve feature selection methods on trajectory inference tasks across two modalities. We highlight the limitations of existing feature selection methods on inferring trajectories from single-cell RNA sequencing data and proteomic imaging data under different technical challenges. For proteomic imaging data, we demonstrate that DELVE outperforms the existing feature selection methods based on its ability to exclude noisy imaging features prior to feature ranking. For single-cell RNA sequencing data, we demonstrate that similarity-based approaches, including DELVE, are more accurate than the commonly used variance-based and RNA velocity approaches. Collectively, this study provides a novel method that can be used for robust trajectory analysis across data modalities, while also providing practical guidelines on feature selection method performance for trajectory analysis.

3. For Eq 5, a p-value is still helpful.

Based on the reviewer's suggestion, we have modified the code to additionally return the permutation p-value when detecting dynamic modules (i.e. *delve._run_cluster* function).

Reviewer #3 (Remarks to the Author):

All my comments are addressed.